# Bio-inspired multimodal soft actuator with environmental self-adaptation

Chi Chen [1], Zixiao Liu [1], Pengju Shi[1], Yusen Zhao[1], Sidi Duan[1], Yingjie Du[1], Yichen Yan[1], Muqing Si[1], Tetsuya Iwasaki[2] & Ximin He [1] ✉

Autonomous soft robots with built-in feedback can achieve self-sustained motion under constant, untethered stimuli. However, these systems are constrained to single-mode actuation within a fixed structure under the same type of stimulus and require human intervention to switch modes, lacking the robust and efficient self-adaptation of living organisms in changing environments. Inspired by *Gymnura micrura*, we developed a light-responsive soft actuator with engineered asymmetry and a dynamic structure, integrating two distinct built-in feedback mechanisms governed by intrinsic bifurcation. Thus, the actuator can seamlessly switch between three different motion modes—tracking, undulation, and oscillation—exhibiting self-adaptation to environmental changes (e.g., light intensity, viscosity, temperatures, and physical contact). Furthermore, this multimodal capability facilitates unique environmental interactions, expanding applications beyond locomotion to include fluid dynamics, electronics, and environmental monitoring. Such an advancement in physical intelligence represents a pivotal step toward next-generation autonomous soft robotic systems, unlocking higher-level autonomy and unprecedented adaptive behaviors.

In recent years, there has been an emerging interest in designing bionic robots capable of adapting to diverse environments by performing appropriate motions, including burrowing robot[1], amphibious robotic turtle[2], and bipedal robot[3]. This self-adaptation trait can minimize the occurrence of uncoordinated changes to increase survivability and reduce energy consumption during operation. To achieve such bio-level intelligence, conventional chip-based robots mimic the multimodal behaviors through complex multi-component hierarchical systems with separate roles, typically including centralized control units for switching motion modes, along with actuator, sensor, and tethered power supply units (Supplementary Fig. 3A). However, this approach inevitably leads to tradeoffs of increasing mass, size, and exponentially growing complexity in systems, which become particularly critical for the development of miniaturized and lightweight robots[4]. In contrast, stimuli-responsive materials, which can change physical properties upon environmental cues, offer an alternative solution to address these issues at the material level by utilizing physical intelligence (Supplementary Fig. 3B)[5–7]. For instance, photo-driven oscillators can simultaneously integrate sensing, decision-making, and actuation in response to the untethered input, leveraging built-in feedback loops for self-sustained motion[8–10]. This eliminates the need for additional control units and overcomes the connectivity limitations of tethered power, presenting the potential for long-lasting, self-sustained robotic systems[11]. Despite these advancements, existing examples of such physical intelligence-enabled soft robots remain in their infancy, constrained to single-mode actuation for self-sustained motion within a fixed structure when triggered by the same type of stimulus, and still dependent on human intervention (e.g., reshaping the robot) to switch motion modes, as their response logic is mechanically pre-encoded into the material during fabrication[12–14]. Therefore, it remains a formidable challenge to further improve the autonomy of soft robotics to enable self-adaptive motion transitions in response to varying conditions (e.g., fluid viscosity, surrounding temperature, and physical constraint)[15].

[1]Department of Material Science and Engineering, University of California, Los Angeles, CA, USA. [2]Department of Mechanical and Aerospace Engineering, University of California, Los Angeles, CA, USA. ✉e-mail: ximinhe@ucla.edu

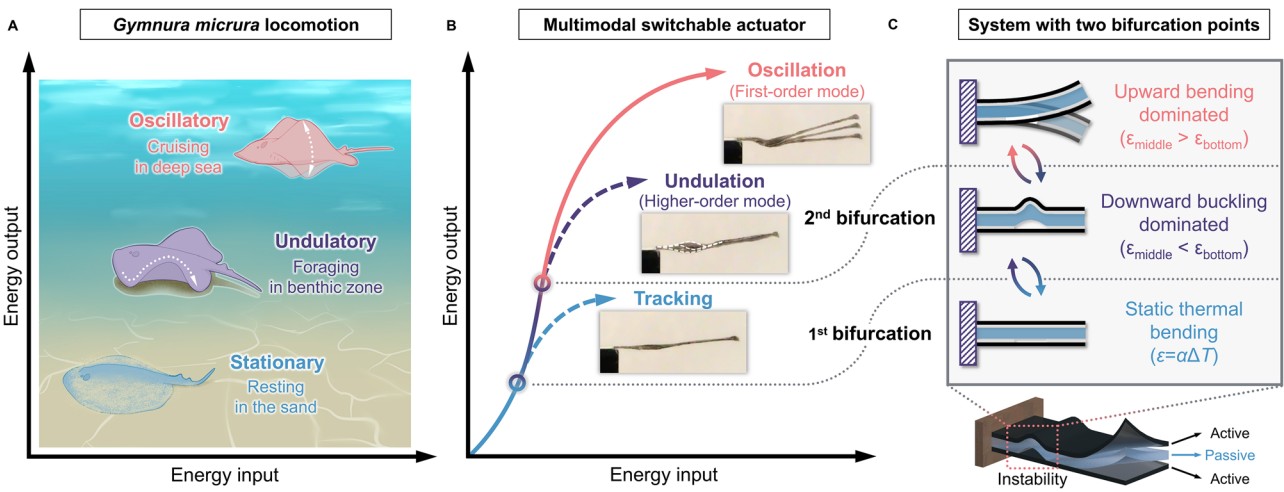

**Fig. 1 | Bio-inspired multimodal switchable actuator with environmental self-adaptation. A** The environmental adaptation of *Gymnura micrura* by switching between multiple motion modes. **B** Multimodal switchability of our soft actuator regulated by different energy states. **C** The design of the nanocomposite system with two bifurcation points, discretizing different motion modes.

Sea creatures inspire us with their ability to seamlessly switch between swimming modes to effectively adapt to diverse environmental conditions[16]. One example is the *Gymnura micrura*, a stingray species that can employ both undulatory (wave number ≥ 1 and phase lag ≥ 360°) and oscillatory (wave number < 1 and phase lag < 360°) locomotion by maneuvering its pectoral fin to swim at different energy outputs (Fig. 1A)[17,18]. When residing in the benthic zone for foraging or reproduction, this species utilizes undulatory behavior with a low-energy state, which is adopted for efficient low-speed swimming while maintaining a high level of control and dexterity. By contrast, during deep-sea cruising, *Gymnura micrura* switches to oscillatory locomotion with a high-energy state, which allows for greater thrust generation at the cost of maneuverability[19]. Similarly, in the realm of structural dynamics, the distinction between undulation and oscillation lies in phase relationships at different points. Oscillation refers to the first mode of motion, characterized by a uniform movement across the entire structure, where all parts experience similar phases of motion. In contrast, undulatory motion corresponds to the higher mode of motion, where different parts of the structure move out of phase with each other, exhibiting more complex behavior with shorter periods and multiple peaks/valleys[20]. Currently, stimuli-responsive polymers have been used to recreate the above cyclic motions through built-in feedback loops that couple deformation with the absorption of energy input (Supplementary Fig. 4A)[21,22]. However, the emergence of a specific motion mode is contingent upon structural dimensions, mechanical properties, the nature of stimuli, and boundary conditions[23]. For example, oscillation is realized by flexible cantilever structures with certain bending stiffness, where localized light exposure induces local photothermal contraction and bending of the cantilever beam, with its tip overshooting to block the light due to inertia, thereby enabling cooling and recovery. Such material-stimulus interaction presents a self-shadowing mechanism and thus forms a built-in feedback loop within the material system (Supplementary Fig. 4B)[24,25]. On the other hand, self-sustained undulation typically relies upon a bow-shaped structure that constrains internal forces, governed by a different built-in feedback loop, where the photothermal crest moves away from the horizontal light until it halts at the clamped end, enabling a new crest to emerge from the other side through a snap-through transition (Supplementary Fig. 4C)[21,26]. Both modes serve important functions in biological kinematics and have been accomplished using a monolithic material system, but each necessitates a distinct predesigned structure, with human intervention needed for switching or reconfiguration[27,28]. To date, switching between different orders of motion modes (e.g., undulation and oscillation) within a single structure under the same kind of stimulus remains unrealized—a key challenge in advancing physical intelligence—due to the difficulty of dynamically modulating the inherent constraint of the structure.

Herein, we propose a self-adaptive multimodal actuator composed entirely of stimuli-responsive soft materials, capable of dynamically switching motion modes according to its energy state (Fig. 1B). Specifically, by introducing a controllable structural asymmetry via creating a defect into an otherwise symmetric photoactive-passive-photoactive trilayer, a stress asymmetry will be produced under constant photothermal input, resulting in a system with two bifurcation points (Fig. 1C). As system energy increases, the internal stress generated at the defect will be large enough to create a transient bow-shaped structure for undulation, where downward buckling dominates, defined as the first bifurcation (tracking-undulation bifurcation). At a higher energy state where upward bending dominates, a second bifurcation will occur as the system switches from higher-order mode (undulation) to first-order mode (oscillation). The availability of motion switching creates a dynamic photo-thermo-mechanical feedback system with the potential for multimodal adaptation under varying conditions (e.g., light intensity, fluid viscosity, surrounding temperature, and physical constraint) controlled by the energy state. This leads to distinct performance outputs—analogous to the behavioral strategy of *Gymnura micrura*—with higher peak output during undulation but higher average output during oscillation, allowing for continuous locomotion across diverse environments. In addition to soft robotic kinematics, this generalized design for generating different environmental interactions can be extended to broader areas, including fluid dynamics, electronics, and environmental monitoring.

## Results

### Tracking-undulation bifurcation

To realize photo-driven autonomous motion with intrinsic photo-thermo-mechanical feedback[29,30], we designed a trilayer system consisting of a polydimethylsiloxane (PDMS) passive layer sandwiched between two photoactive liquid crystal elastomer (LCE) layers doped with candle soot nanoparticles (CSNPs) (Supplementary Fig. 5). Due to the homogenous dispersion, high photothermal efficiency, and exceptional compatibility of CSNPs with LCE that prevents the aggregation or precipitation[31], we achieved an optimal balance between the actuation performance and mechanical stiffness of the LCE/CSNP active layer[28,32]. To engineer the multiple bifurcation kinematics, we intentionally introduced an unbound region between the passive

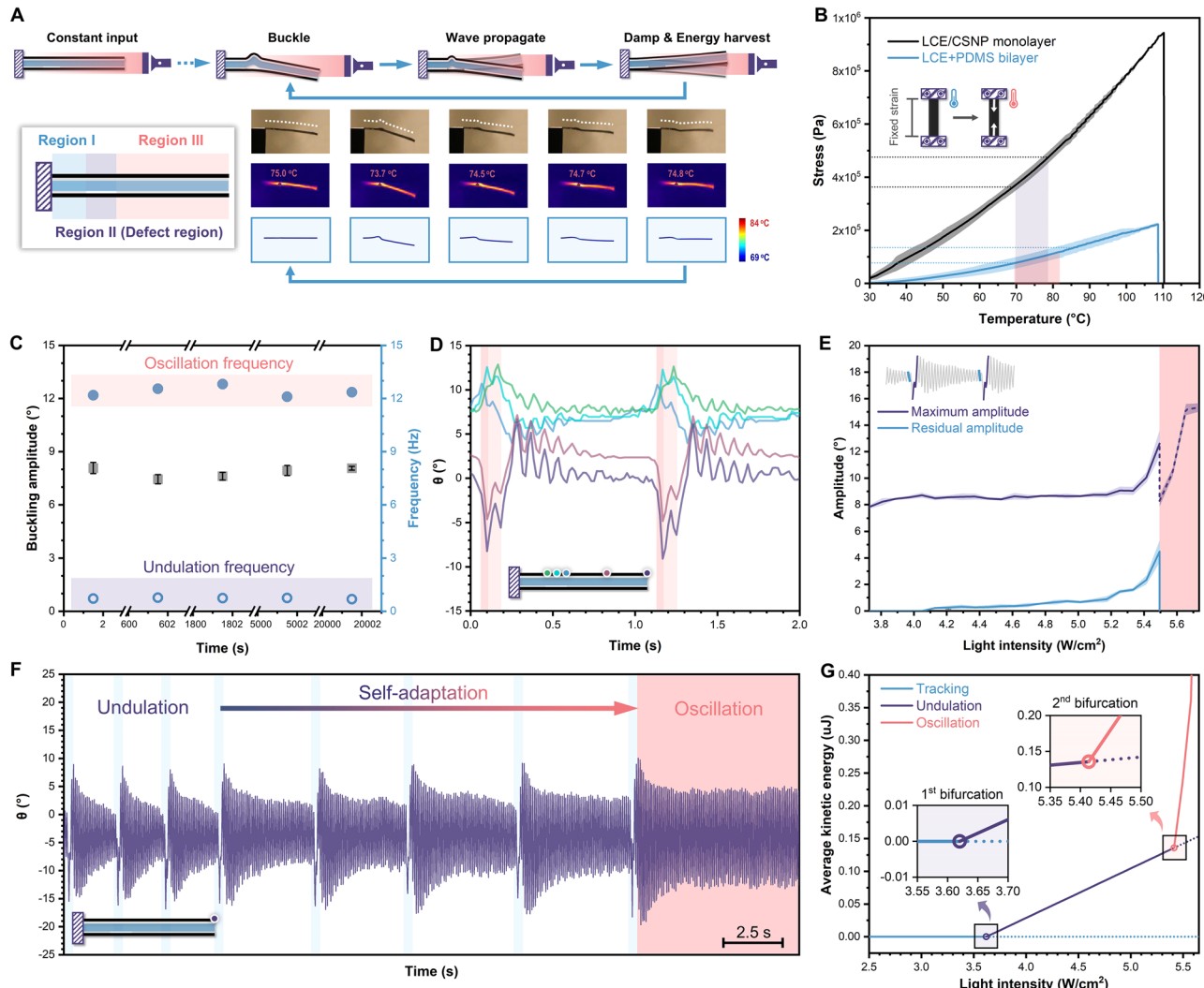

**Fig. 2 | Instability-induced undulation and bifurcation. A** Schematics and snapshots of the asymmetric trilayer during the undulation cycle from optical images, IR images, and simulation results. **B** Actuation stress of LCE/CSNP monolayer and bilayer under DMA isostrain condition (pink and purple regions refer to the temperature ranges of LCE/CSNP monolayer and bilayer, respectively, under constant illumination). **C** Comparison of tip amplitude and frequency over a long time, where the undulation frequency is defined as the buckling frequency. **D** Motion tracking with the angle change during the first two cycles at the input of 3.728 W/cm² (green, cyan, and blue refer to left, middle, and right points of the defect region, while violet and purple refer to the midpoint and tip of Region III, respectively). **E** Amplitude adaptation to increasing input, calculated from the trajectory of the cantilever tip (white region for undulation, while pink region for oscillation; the maximum amplitude is the buckling amplitude in undulation or the oscillation amplitude in oscillation; the residual amplitude is the amplitude of the last complete swing before buckling in undulation, which does not exist in oscillation). **F** The undulation-oscillation bifurcation at the input of 5.494 W/cm². **G** Average kinetic energy variation across the two bifurcation points.

PDMS and bottom active LCE/CSNP layer to create structural instability in the trilayer system, forming an asymmetric structure with an artificial defect. This engineered localized delamination between the two layers was observed under scanning electron microscopy (SEM) (Supplementary Fig. 6). Therefore, we can divide the entire structure into three parts, herein termed as: Region I, Region II (defect region), and Region III (Fig. 2A, insert). The asymmetric defect region, Region II, consists of a bilayer and an LCE/CSNP monolayer with distinct thermomechanical properties (Supplementary Fig. 7). As a result, when the trilayer was first exposed to constant near-infrared (NIR) illumination from the unconstrained side, the top bilayer reached a higher temperature (69.7–82.3 °C) compared to the bottom LCE/CSNP monolayer (69.1–78.7 °C) because the tip of the trilayer naturally dipped downward due to gravity (Supplementary Fig. 8A). Thermal energy accumulated on the outer active layers regardless of where the light was focused and would eventually be transferred along the structure to the defect region. In Region II, although the bilayer

exhibited a higher temperature under constant illumination, it generated less than one-third of the stress (77.1–134.9 kPa) compared to the LCE/CSNP monolayer (362.9–475.6 kPa), as indicated by the DMA results (Fig. 2B). Such antithetical trends between temperature and stress constituted the instability for bifurcation.

At lower light intensities below the tracking-undulation bifurcation point (3.728 W/cm²), there was an inadequate difference in stress, leading to static thermal bending for tracking behavior towards the irradiation. A larger stress difference amplified the instability and led to the buckling of the top bilayer to generate a transient bow-shaped structure. In this scenario, the constant irradiation initiated a traveling wave within the buckled Region II, from right to left (Supplementary Fig. 9 and Movie 1)[21]. As a result, tracking-undulation bifurcation from stable to unstable state was realized[33]. Meanwhile, Region III bent upwards to achieve the self-shadowing mechanism[22]. IR camera imaging and simulation modeling revealed a sudden temperature drop at each buckling moment, followed by a gradual temperature rise that

initiates the next cycle (Fig. 2A right and Supplementary Fig. 8B). To quantitatively analyze the undulation, five points selected along the trilayer were tracked. During a continuous period of 20,000 seconds under identical irradiation, every waveform in Region II and III remained constant without obvious attenuation and shared a similar buckling amplitude and frequency (Fig. 2C, Supplementary Fig. 10, and Movie 1). More importantly, there existed an apparent phase difference between different positions in the defect region (-10.5° between green and cyan points while ~15.6° between cyan and blue points) (Fig. 2D), indicating strong evidence for the instability-induced undulation. Meanwhile, the selected points in Region III remained in the same phase, characteristic of damping oscillation, which would not interfere with the asymmetric structure. Afterward, we turned the trilayer upside down and observed the similar continuous undulation, with the wave propagating along the structure from the unconstrained side, while the tip was even able to buckle against the direction of gravity (Supplementary Fig. 11). This suggests that gravity is not the dominant factor in this instability-induced behavior; instead, the primary contributors are the structural asymmetry and the photo-thermo-mechanical coupling.

## Undulation-oscillation bifurcation

The change in light intensity, as the only energy source, plays a decisive role in the motion output, determining both undulation amplitude and frequency. Starting under relatively low input (3.728–4.049 W/cm²), the whole structure was damped to an equilibrium tracking state (residual amplitude = 0) (Fig. 2E and Supplementary Movie 2). In this scenario, while oscillation of Region III could be initiated by periodic undulation of Region II, it was eventually damped out before the next cycle of undulation. The rationale is that the input energy was partially released to buckling, while the remaining thermal energy temporarily fueled the damped oscillation. As the input energy damped out gradually, the light illuminated the top bilayer to trigger the next cycle. When the energy input increased incrementally, the residual amplitudes rose, gradually supporting sustainable oscillation within every undulation cycle (Supplementary Fig. 12). Notably, further increasing the light intensity to the undulation-oscillation bifurcation point (5.494 W/cm²), the motion spontaneously switched to complete oscillation without any buckling or undulation (Fig. 2F, Supplementary Fig. 13 and Movie 2)[19]. Such bifurcation is a result of competition between the bending and buckling tendency of the defect region, as modeled below

$$\text{At middle/bottom interface}$$
$$\varepsilon_2 = \alpha_1 \Delta T_{12} + \frac{P_1}{a_1 E_1} - \frac{a_1}{2\rho} - \frac{a_2}{\rho}, \ \varepsilon_3 = \alpha_3 \Delta T_{23} + \frac{L-d}{L}\frac{P_3}{a_3 E_3} + \frac{a_3}{2\rho} \quad (1)$$
$$\varepsilon_2 < \varepsilon_3, \text{ undulation}; \ \varepsilon_2 > \varepsilon_3, \text{ oscillation}$$

This equation reveals three parts contribute to the contraction strain of the middle PDMS layer ($\varepsilon_2$) and the bottom LCE/CSNP layer ($\varepsilon_3$): thermal strain, axial elastic strain, and bending strain. Specifically, the higher light intensity heats the top surface, generating a larger thermal strain $\alpha_1 \Delta T$, where $\alpha_1$ represents the negative coefficient of thermal expansion of LCE/CSNP, thereby creating a tendency to bend upward. Due to the thermal diffusivity ($k/\rho C_p$), only a limited amount of heat can be transferred to the bottom layer. Meanwhile, the unbonded defect region in the bottom layer experiences reduced geometric and mechanical constraints, leading to increased deformability under irradiation (Fig. 2B). Thus, despite receiving lower heat and temperature compared to the top surface, the bottom layer is still capable of bending down, leading to a buckling tendency. At the undulation-oscillation bifurcation point, the bending curvature of the top layer, $1/\rho = M_1/E_1 I_1$ (Eqn. S3), changes from negative to positive, signifying the predominance of the bending tendency over the previously dominant buckling tendency. Consequently, at the middle/bottom layer interface, $\varepsilon_2$ is greater than $\varepsilon_3$ [Eq. (1)], resulting in the

crossing of the undulation-oscillation bifurcation point for motion switching (Supplementary Text 2.1).

During this bifurcation, the buckling phenomenon vanished and the residual amplitude merged with the maximum amplitude. The cyclic motion before the undulation-oscillation bifurcation can be decoupled into undulation and a damping oscillation, each with its own frequency that depends solely on material properties and is invariant against light intensity change (Supplementary Fig. 14). After entering the oscillation regime, higher input increased the amplitude and frequency to improve the average kinetic energy rather than change the motion mode (Supplementary Figs. 13–15). Upon conducting the modeling analysis, it was observed that the average kinetic energy demonstrated a remarkably larger slope at two bifurcation points (Fig. 2G). Besides, when the energy input was lowered below the undulation-oscillation bifurcation point, the trilayer self-regulated back to undulation (Supplementary Fig. 16 and Movie 3). Simply, such an asymmetric structure can spontaneously select and self-adjust its motion based on the current energy input.

## Governing factors of bifurcation

The geometrical parameters of the asymmetric structure determine the resulting modes of motion. Herein, we adopted the thickness ratio of active:passive:active layers as 1:2:1, and the width would not influence the motion behavior as long as the irradiation width can cover the trilayer[32]. Thus, the key parameters can be simplified to $X/L$ (the normalized length of Region I) and $d/L$ (the normalized length of Region II). The experimental results were presented in Fig. 3A, where all samples demonstrated the tracking ability but the other two motion modes were more sensitive to changes in $X/L$ and $d/L$. For instance, when $X/L$ was fixed at 0.20, undulation could be observed with low $d/L$ (Supplementary Fig. 17A). Increasing $d/L$ resulted in the enhancement of both the maximum amplitude and the residual amplitude, since the larger length led to a more pronounced energy imbalance to generate buckling (Fig. 3B). However, only the range $0.10 \le d/L \le 0.12$ successfully achieved the undulation-oscillation bifurcation, while delamination occurred in trilayers with $d/L > 0.12$ due to excessive energy exceeding the adhesive limit and destroying structural integrity. Moreover, both larger $X/L$ and $d/L$ lowered the triggering light intensity for tracking-undulation bifurcation. On the other hand, when $d/L$ was fixed at 0.12, bifurcation occurred only within the range $0.20 \le X/L \le 0.30$, with the resulting undulations exhibiting similar amplitudes and frequencies (Fig. 3C and Supplementary Fig. 17B). This indicated that the position of the defect region primarily determined whether a tracking-undulation bifurcation occurred, rather than simply influencing the undulation characteristics such as amplitude or frequency.

The effects of $d/L$ and $X/L$ were also considered in the context of our undulation-oscillation bifurcation model. Specifically, increasing $d/L$ produces a larger elastic strain $(L-d)P_3/a_3 E_3 L$ in the bottom layer [Eq. (1)]. The effects of $X/L$ are not explicitly shown in the equation, but $X/L$ affects the relative position between the illuminated and unbound regions, further influencing the temperature and thermal strain (e.g., $\alpha_3 \Delta T_{23}$ and $\alpha_1 \Delta T_{12}$ in our analytical model). As a result, if $X/L$ is too small or too large, the symmetric region, where the curvature of the three layers is always the same, will be illuminated without undulation. For a given $d/L$ and $X/L$, the undulation-oscillation bifurcation criterion in Eq. (1) can be simplified by defining a ratio $K_0$ between the thermal strain of LCE/CSNP at the middle/bottom interface $\alpha_3 \Delta T_{23}$ and the thermal strain at the top/middle interface $\alpha_1 \Delta T_{12}$ in Eq. (2). Since the top and bottom layers are fabricated from the same LCE/CSNP recipe under identical synthesis conditions (i.e., $\alpha_1 = \alpha_3 = \alpha_{LCE/CSNP}$), these terms cancel out in Eq. (2). Thus, we can obtain a $K_0$ map for given geometrical parameters, Fig. 3D (see details in Supplementary Text 2.2). Note that the $K_0$ represents the theoretical value of the bifurcation point, and is required to be compared with the experimentally measured or

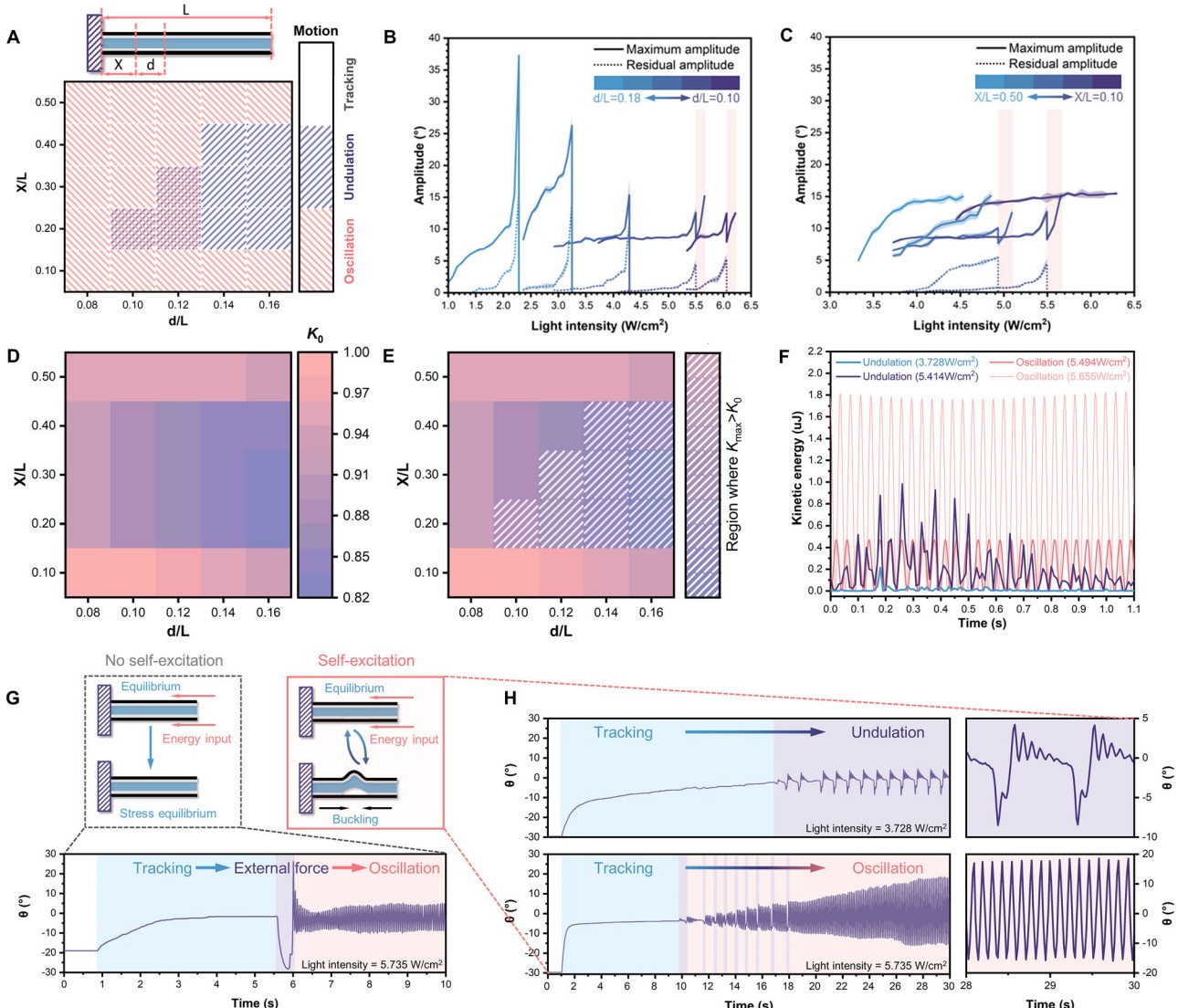

**Fig. 3 | Regulation of bifurcation and self-excitation behavior. A** Influence of structural parameters on multimodal motion ($X$ and $d$ refer to the length of Region I and Region II, respectively). **B** Comparison of motions with different values of $d/L$ when $X/L$ was fixed to 0.20. **C** Comparison of motions with different values of $X/L$ when $d/L$ was fixed to 0.12. **D** Simulation results of theoretical $K_O$ for the undulation–oscillation bifurcation. **E** Comparison of numerically simulated $K$ and $K_O$ (white region refers to where the undulation can occur). **F** Comparison of kinetic energy between undulations and oscillations. **G** Excitation comparison of the symmetric and asymmetric trilayer structures. **H** Self-excitation of the asymmetric structure with different inputs.

numerically simulated thermal strain ratio, denoted as $K$, to determine the state of the system. Such numerically simulated results update the map to Fig. 3E, where the white region indicates the occurrence of undulation—the key condition for achieving two bifurcations overserved in experiments (Fig. 3A).

$$K_0 = \frac{\alpha_3 \Delta T_{23}}{\alpha_1 \Delta T_{12}} = \frac{\alpha_{\text{LCE/CSNP}}(T_{23} - T_0)}{\alpha_{\text{LCE/CSNP}}(T_{12} - T_0)}, \begin{cases} K > K_0, \text{ undulation} \\ K < K_0, \text{ oscillation} \end{cases} \quad (2)$$

For example, for $X/L = 0.20$ and $d/L = 0.12$, the theoretical bifurcation point $K_0$ is 0.8700, whereas the numerically solved thermal strain ratio $K$ is 0.8705 at 3.728 W/cm², which suggests the onset of undulation (Supplementary Fig. 18). Furthermore, modeling reveals bursts of energy output during each cycle in the undulation mode, whereas the oscillation mode yields a higher average kinetic energy output despite exhibiting lower energy maxima near the undulation–oscillation bifurcation point (Fig. 3F and Supplementary Text 2.3). Specifically, the peak output efficiency of undulation is more than twice that of oscillation, whereas its average output

efficiency is only 0.681 times that of oscillation (Supplementary Fig. 19). This trade-off is akin to the behavioral strategy of *Gymnura micrura*, which employs undulatory swimming at lower speeds to enhance instantaneous output and maneuverability during predation or reproduction, and shifts to oscillatory swimming for high-speed cruising[19].

## Self-excitation

The introduction of the defect is also crucial for facilitating a self-excitation behavior due to the associated decrease in structural stiffness. In contrast, forced triggering was necessary in the case of the trilayer structure without engineered asymmetry (Fig. 3G and Supplementary Movie 4). Since the motion switching was only based on energy inputs, the system would autonomously adapt to the proper motion mode and skip the intermediate state when necessary, e.g., from tracking directly to oscillation (Fig. 3H and Supplementary Movie 5). Additionally, when we introduced a second laser with additional energy to the asymmetric structure in the equilibrium tracking state, it would self-regulate to the higher energy state and perform

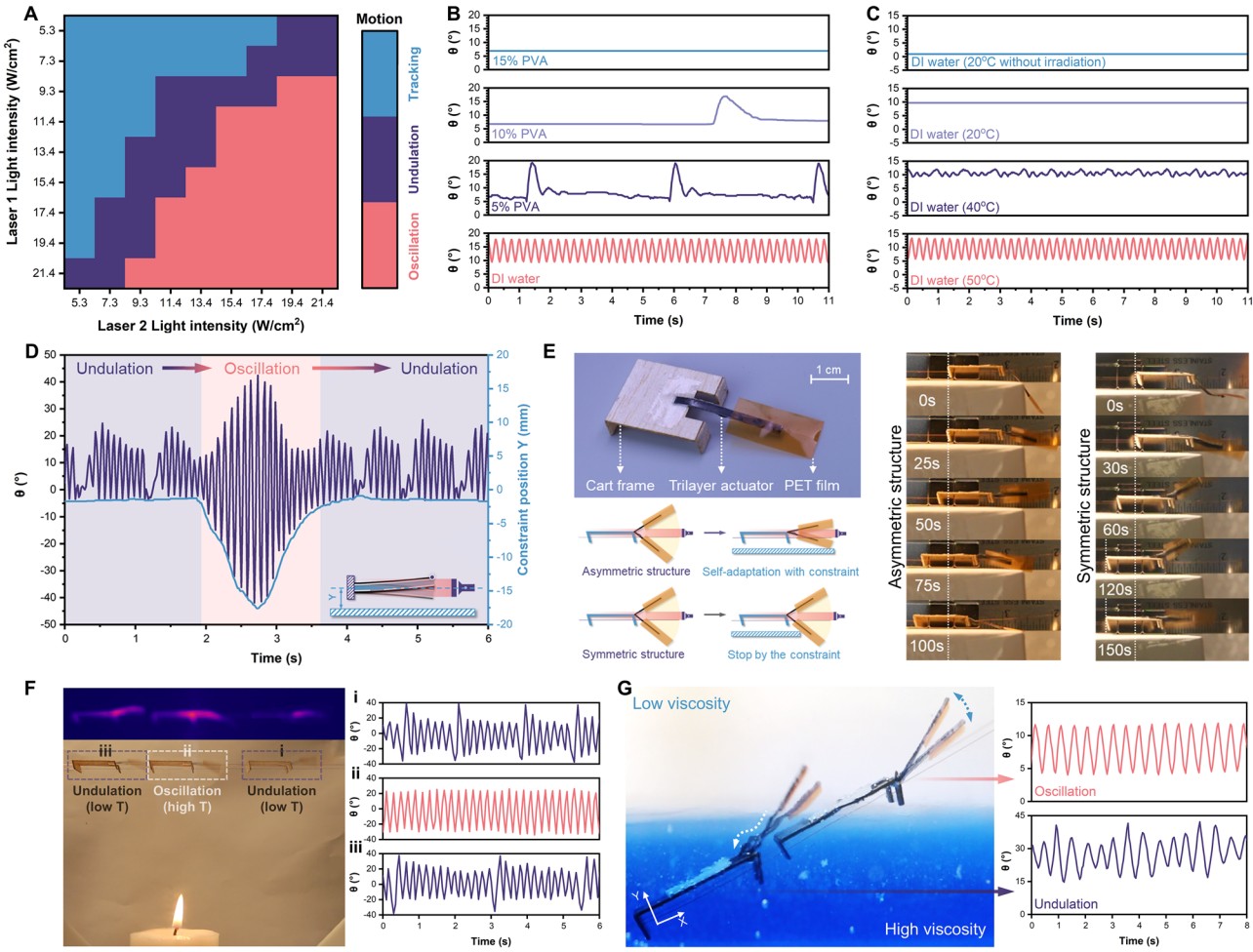

**Fig. 4 | Self-adaptation with varying environments.** Motion switchability under fluid conditions with varying **A** light intensity, **B** viscosity, and **C** temperature. **D** Self-adaptation to the physical constraint. **E** Cart design for self-continuous locomotion with asymmetric trilayer, while the symmetric counterpart was blocked by the physical constraint. **F** Self-adaptation of the cart passing through the high-temperature zone (starting from i, passing through ii, and ending in iii). **G** Self-adaptation of the cart passing through the liquid-liquid interface with different viscosities.

undulation, regardless of the laser incidence direction (Supplementary Fig. 20 and Movie 6).

### Advancing physical intelligence in varying environments

Since bifurcation points were invariant for one prepared structure, the corresponding motion could be tuned by changing the environmental dissipation under a constant input. Firstly, we demonstrated that this trilayer structure can replicate the kinematics of *Gymnura micrura* under aquatic conditions. The results confirmed that the energy law could be extended to aqueous environments. That is, the motion of the asymmetric structure is determined by the total energy accumulated regardless of the input incidence (Fig. 4A, Supplementary Fig. 21A, and Movie 7). By tuning the viscosity of the solution through varying polyvinyl alcohol (PVA) concentrations, the structure spontaneously switched motion modes, effectively adapting to environmental changes (Supplementary Fig. 21B–D). Specifically, increasing viscosity with higher PVA concentration led to greater resistance and more energy dissipation, causing a decrease in system energy and thereby the motion transition from oscillation to undulation, and eventually to static tracking (Fig. 4B and Supplementary Movie 8). Similarly, the self-adaptation between various motion modes could be controlled by adjusting the surrounding temperature to affect energy dissipation through the thermal conduction rate (Fig. 4C and Supplementary Movie 8).

When physical contact was introduced into the system, a portion of the kinetic energy was dissipated through collision, leading to

a reduction in the system energy and the transition from oscillation to a lower-energy undulation state (Fig. 4D, Supplementary Fig. 22, and Movie 9). To demonstrate its application, we first integrated the asymmetric trilayer as an actuator with a PET film into a cart frame made of Balsa wood, which was perforated and suspended on the copper wires to guide the locomotion direction (Fig. 4E). Under constant irradiation and physical contact with the ground, the asymmetric actuator could continuously convert light energy into mechanical actuation and generate powerful stroke to drive the cart forward. Moreover, it would self-regulate its gait to ensure continuous forward movement once in contact with the ground (Supplementary Fig. 23A and Movie 10). In contrast, the symmetric counterpart was impeded by the same constraint even under significantly higher input (Supplementary Fig. 23B and Movie 11), due to the lower kinetic energy bursts that were insufficient to overcome the friction (Fig. 3F). This undulation of the asymmetric actuator indicated its ability to achieve the higher maneuverability to navigate obstacles under lower energy output, paralleling the undulatory locomotion of *Gymnura micrura* that adopts this strategy at low speeds to improve controllability in confined spaces. Furthermore, when the cart passed through a high-temperature zone, the motion mode of the asymmetric actuator alternated between undulation and oscillation, adjusting to the fluctuating temperatures without human interference (Fig. 4F, Supplementary Fig. 24 and Movie 12). Similarly, as the cart traversed the liquid-liquid interface with different

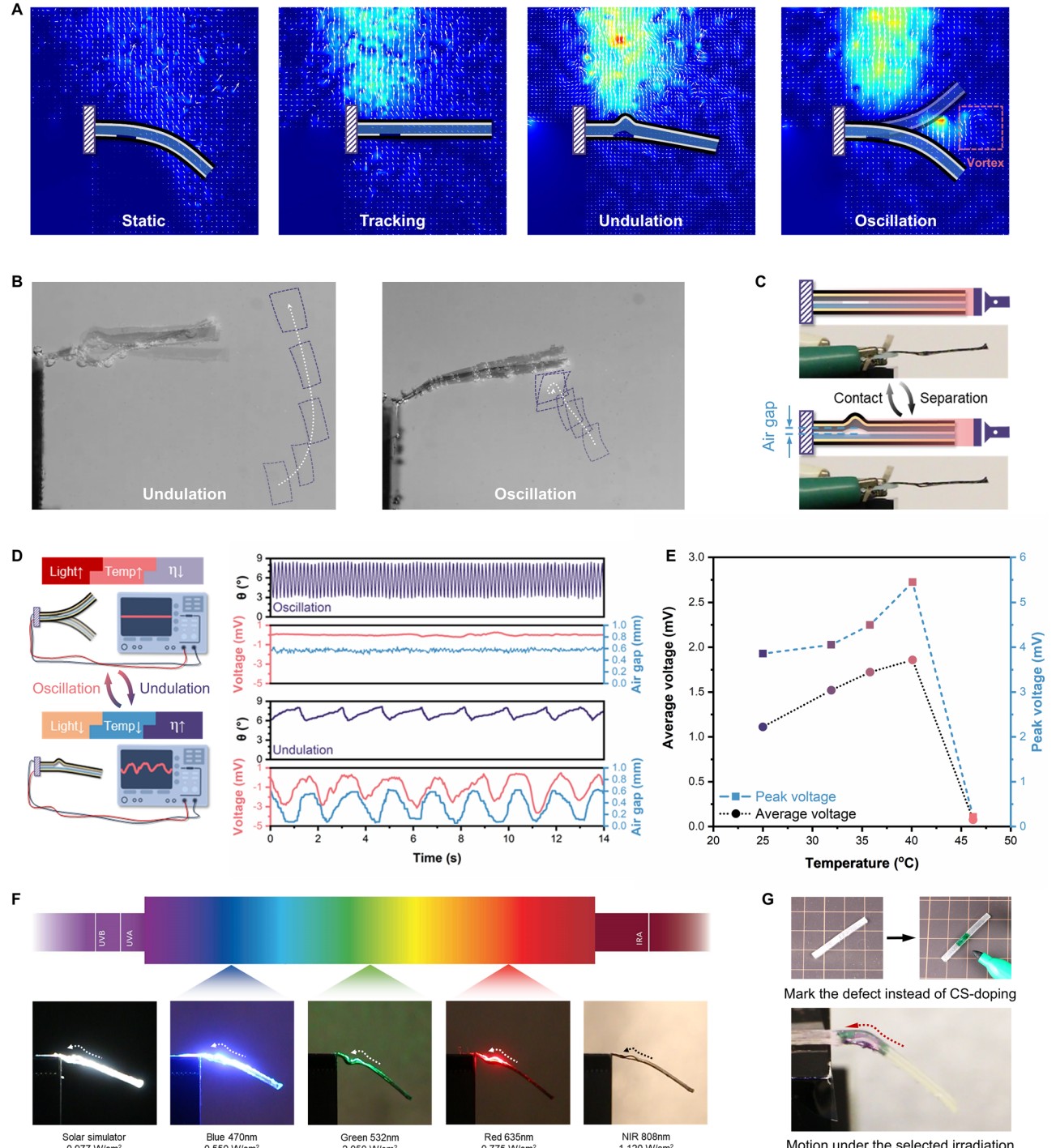

**Fig. 5 | Multifunctionality and generalization. A** PIV analysis with enhanced airflow during undulation and the formation of a vortex during oscillation. **B** Different aquatic interactions between the asymmetric actuator and particles during undulation and oscillation. **C** Contact-separation mechanism during undulation for cyclic voltage output. **D** Comparison of TENG voltage outputs between undulation and oscillation under varying conditions. **E** Temperature-dependent variation of TENG voltage output. **F** Broad-spectrum operation of the asymmetric actuator. **G** The selective and erasable coating for the asymmetric actuator.

viscosities (Supplementary Fig. 25), its motion mode changed accordingly (Fig. 4G and Supplementary Movie 13).

## Multifunctionality and generalization

Moreover, the undulation-oscillation bifurcation entails a distinct change in the surrounding airflow pattern (Fig. 5A, Supplementary Fig. 26, and Movie 14). Particle image velocimetry (PIV) demonstrated that the buckling during undulation enhanced the updraft airflow over the defect region, while the oscillating actuator led to a unique vortex

around the tip area (Supplementary Fig. 27). This is analogous to the non-reciprocal vortex existing in oscillating fishtails that generates thrust, which highlights the potential of our asymmetric actuator for fluid dynamics applications, including fluidic transportation and particle trapping during oscillation in fluid environments (Fig. 5B)[16,24]. Additionally, the periodic vacancy changes in the bow-shaped structure during buckling-induced undulation make the asymmetric actuator promising for developing triboelectric nanogenerators (TENG) by adopting PTFE/Pt and PDMS/Pt as inner layers

(Supplementary Fig. 28). In the undulation mode, the positive and negative triboelectric layers created the contact-separation cycle to generate the open-circuit voltage with every solid-solid contact (Fig. 5C), which would not occur in the oscillation state due to insufficient change in the air gap (Fig. 5D)[34,35]. This distinct bifurcation can be applied not only to monitor specific environmental changes—such as voltage drops indicating viscosity or temperature crossing the threshold—but also to control desired voltage outputs (both average and peak voltage output) by manually tuning environmental conditions (Fig. 5E).

Lastly, all the adaptive performances above could be triggered remotely across a range of wavelengths since CSNP endowed the system with a broad absorption spectrum spanning from visible to near-infrared regions (Fig. 5F and Supplementary Fig. 29)[28,36]. Notably, motion initiation could also be achieved by collimated light, apart from localized lasers, as long as the irradiation covered the defect region with appropriate intensity. Furthermore, the multimodal behavior could be reproduced by marking the defect region with erasable commercial light-absorbing inks instead of CSNP doping (Fig. 5G and Supplementary Fig. 30). Meanwhile, we tried to extend the LCE/PDMS system to more material systems (Table S1). Among several commercial materials tested as interlayer replacements, nitrile demonstrated similar buckling behavior and exhibited the two bifurcation points (Supplementary Fig. 31), attributed to its comparable thermomechanical properties (Table S2 and Supplementary Fig. 32). Besides, the active-passive-active structure was essential for instability-induced motions, because the outer layers generated imbalanced stress while the interlayer conducted heat and provided structural support (Table S3). Moreover, we attempted to introduce additional defects into Region III and found that the presence of a second defect could influence all three motion modes and induce an additional oscillation frequency in the oscillation mode (Supplementary Fig. 33 and Movie S15). Overall, we believe that such multimodal adaptation can be extended to more material systems with proper thermo-mechanical properties and comparable designs.

## Discussion

We have proposed an auto-switchable soft actuator design that enables multiple self-sustained motion modes with built-in feedback across diverse environments. This is achieved by introducing a controllable defect into an otherwise symmetric trilayer and harnessing photo-thermo-mechanical dynamics within an entirely soft material system. This simple structure advances the physical intelligence of soft actuators by enabling autonomous regulation of motion modes across two bifurcation points based on the energy state. In principle, the system energy can be determined by (i) remote input (e.g., light intensity) and (ii) environmental dissipation (e.g., fluid viscosity, surrounding temperature, physical contact). Consequently, this generalizable design for self-excitation and motion self-adaptation represents a promising strategy for developing chipless, untethered soft robots with higher-level autonomy and expanded functionality. Furthermore, the combination of undulation and oscillation broadens the scope of current multimodal robotic systems and may inspire the development of other disciplines, such as fluid dynamics, electronics, and environment monitoring.

## Methods

### Materials preparation

The preparation of LCE/CSNP followed our previous routine based on a two-step thiol-acrylate click reaction, detailed in Supplementary Information. The PDMS and Dragon skin films were prepared by mixing the base and curing agent at a ratio of 5:1. After vacuum defoaming for at least 5 min, the mixture was cast into glass molds with predefined spacers and cured at 80 °C overnight. Thin platinum layers, serving as electrodes, were deposited by sputtering (Denton Desk II) on one side

of the PDMS and on one side of the PTFE layer for 45 s and 60 s, respectively.

### Fabrication of the asymmetric trilayer

The prepared LCE/CSNP and PDMS monolayer should be pretreated to enhance the adhesion before the assembly. Specifically, the LCE/CSNP film was cut into the size of 25 mm × 2.5 mm × 140 μm, followed by immersing in ethanol overnight. In the meantime, the PDMS film was cut into the size of 25 mm × 2.5 mm × 280 μm and washed with hexane. Both washed strips were treated with oxygen plasma for 3 min, followed by the deposition of a thin layer of silica coating using chemical vapor deposition with TEOS and ammonia solution for 90 min. Then, two LCE/CSNP strips and one PDMS strip were assembled using silicone glue (10 wt% in hexane) layer by layer from bottom to top (Supplementary Fig 5A). It is important to note that during the assembly of the bottom layer of LCE/CSNP and PDMS, PTFE tape needed to be placed in the predefined position to block the glue and create the defect. Next, the assembled trilayer was transferred to an oven and heated at 60 °C for 3 days to remove hexane. After the PTFE tape was removed, the asymmetric trilayer with an unbound region was obtained (Supplementary Fig. 6). Unless otherwise specified, all tested trilayers have an X of 5 mm (Region I length) and a d of 3 mm (Region II length, same as the width of PTFE tape).

### Motion capture and analysis

All structures were mounted horizontally on a stage and illuminated by an 808 nm laser (LSR808NL, DPSS Laser) from a horizontal direction. The light spot had rectangular dimensions of 3.5 mm × 4.5 mm, which could fully cover the cross-sectional area of the actuator. Besides, a linear translational stage was fixed on the laser to control the precise height of the irradiation towards the sample. All motion was captured and recorded by a camera (EOS 70D, Canon) at 60 fps. The tips or the selected points were tracked by the software Tracker to measure angles and positions, which were subsequently used to calculate frequency and amplitude. Each point with different input was recorded for at least 5 s once the motion had stabilized, followed by analysis of the motion during every one-second period. The results were processed to obtain average frequencies and amplitudes. Actuation under various wavelengths was performed using different light sources including a red 635 nm laser (LSR635NL-100, DPSS Laser), a green 532 nm laser (Genesis MX-532, COHERENT), a blue 470 nm LED driver (SM1U25-A, Thorlabs) equipped with a collimator (M34 × 0.5 EXT, Thorlabs), and a solar simulator (HAL-320W, Asahi Spectra).

### Thermomechanical characterizations

All samples were annealed at 150 °C for 5 min and then cooled to remove any thermal history. Moduli, stresses, and lengths were measured using a dynamic mechanical analyzer (DMA850, TA Instrument) under a 25 kPa preload, 1 Hz oscillation frequency, and a 5 °C/min heating rate. Isostrain mode was employed to measure internal stress and internal force when both ends were fixed without strain variation under 5 °C/min heating rate. The adhesion force was obtained by the peel testing, with two different layers fixed separately in clamps at each end.

### Other characterizations

Differential scanning calorimetry (DSC250, TA Instrument) was performed to measure the nematic-to-isotropic transition of LCE/CSNP and the thermal property of PDMS. The cross-section morphology was captured by a scanning electron microscope (SEM) (Supra 40VP, ZEISS). The CSNP morphology image was obtained using a scanning transmission electron microscope (TEM) (FEI Titan 80-300), with the CSNP dispersed in toluene and the solvent subsequently removed. The photothermal process and thermographic images were captured by an infrared camera (TiX 580, Fluke) and analyzed by the software Fluke

Connect. UV-Vis spectra were measured to investigate the absorption or transmittance using a spectrometer (UV-3101PC, Shimadzu). The particle size of CSNP was measured using dynamic light scattering (DLS) (Zetasizer Pro, Malvern Panalytical), with CSNP dispersed in toluene.

## Setup for aquatic tests

The setup consisted of a tank to hold the sample stage and two NIR lasers to increase input energy, as heat dissipation underwater is significantly greater than in air (Supplementary Fig 21A). The first laser was set horizontally facing the sample, while the second one was aligned at a -15-degree angle to target the defect region. The PVA solution was prepared by dissolving the PVA powder in DI water with rapid stirring at 80 °C, followed by centrifugation in a mixer (ARE-310, Thinky) to remove the bubbles. Viscosity measurements were conducted by a rheometer (Discovery HR 30, TA Instruments) at room temperature under the shear rate from 1 to 100 1/s. The tests under different water temperatures were based on a similar setup by tuning the temperature of the DI water.

## Setup for cart design

Apart from a trilayer as the actuator, the setup consisted of a frame of Balsa wood and a PET film (Fig. 4E). The mass of the actuator was 43.2 mg, while the total mass was 129.8 mg. The four feet of the frame were perforated to allow copper wires to pass through and guide the locomotion direction, which was directly opposite to the incident light. The cart was initially positioned away from the physical constraint to enable the free motion of the actuator. However, the distance between the copper wire and the constraint in the front path was less than the amplitude of the trilayer. For the physical contact demonstration, the light intensity to trigger the asymmetric trilayer was $3.967 \, W/cm^2$, while the intensity for the symmetric trilayer was $6.723 \, W/cm^2$. For the high-temperature passing demonstration, the light intensity was $4.772 \, W/cm^2$. For the liquid-liquid interface passing demonstration, the upper layer was ethanol while the bottom layer was an ethanol solution with 5 wt% D-glucose and blue dye, and the light intensity was $9.348 \, W/cm^2$.

## Particle image velocimetry

First, the tracking particle, BEHS, was added to an atomizer for uniform dispersion in the air. After setting the stage and sample in a sealed acrylic chamber, a 532 nm laser (Genesis MX-532, COHERENT) equipped with a line pattern engineered diffuser was turned on to image the airflow (Supplementary Fig. 26A). The linear irradiation was aligned directly with the center of the sample. When no significant airflow can be observed, the NIR laser was activated to initiate the motion. The entire process was captured by a high-speed camera (Phantom with the 100 mm lens, Laowa) and analyzed by the PIVlab application in Matlab.

## Triboelectric nanogenerator test

The assembled photothermal TENGs were tested by an electrochemical workstation (CH Instruments, CHI660E). The sputtered electrodes were connected to the working and counter electrodes of the electrochemical workstation, while the reference electrode was grounded. The open-circuit voltage was measured using the open-circuit potential-time mode of the electrochemical workstation. The entire setup was placed in a high-temperature oven equipped with a thermocouple to evaluate the output voltage under different temperature conditions.

## Data availability

All data needed to evaluate and support the conclusions in the manuscript are included in the main text or the Supplementary Information. Contact X.H. for any questions.

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

## Acknowledgements

We thank Dr. Wenzhong Yan for his participation in discussions and assistance with equipment setup. This work was supported by the Johnson&Johnson WISTEM2D Scholar award, the American Chemical Society (GR-63402), the Office of Naval Research (ONR) awards (N000142412187 and N000142212595), and the Moore Foundation Award (12072).

## Author contributions

X.H. conceived the concept and planned the project. C.C. designed and conducted the experiments and data analysis. Z.L. developed the simulation with the model and theoretical analysis. C.C. and P.S. conducted the fabrication and characterization. C.C. and M.S. developed the demonstration. C.C. and S.D. drew the schematic diagram. T.I. guided the discussion of the control theory. All authors wrote the manuscript under the supervision and participation of X.H. All authors have given approval to the final version of the paper.

## Competing interests

The authors declare no competing interests.
