## [Transparent Peer Review file · Nature Communications]

Bio-inspired multimodal soft actuator with environmental self-adaptation

Corresponding Author: Professor Ximin He

Version 0:

Reviewer comments:

Reviewer #1

(Remarks to the Author)

In this manuscript, Chen et al. developed a light-responsive soft actuator with intrinsic geometric asymmetry (debonding defects in trilayer composite). Three modes of motion are identified: namely tracking, undulation, and oscillation. Through systematic experimental studies, the authors explored the optimal geometric parameters to allow the soft actuator to alter between three different modes upon varying input light intensity from the laser beam. The authors also demonstrated that motion modes can be switched based on environmental variation such as the changing of viscosity, temperatures, and physical contact. The authors showed some robotic functions based on its environmental adaptivity such as the soft robot can pass through the high-temperature area and swim from a low-viscosity area to a high-viscosity area. Lastly, the authors demonstrated the multifunction of such light-responsive soft actuator based on different motion modes in altering the airflow patterns, TENG. The light-responsive soft actuator can also work under different spectrums of laser input and the defect region can be replaced with more commercially available materials.

The manuscript is well written and details of the design of the soft actuator are given. Such mode switch of light-driven soft actuator is sound. The mechanism governing the bifurcation phenomenon is well explained. It is an inspiring work and can attract attention from experts working in the soft robotics. However, the reviewer has several major comments on this work: First, While the presented soft robotic actuator demonstrate self-regulating and adaptive behaviors—switching between undulatory and oscillatory modes in response to energy states—characterizing it as “life-like intelligent” can be somewhat misleading. Fundamentally, the system relies on built-in mechanical feedback loops facilitated by material properties passively (e.g., photothermal effects, stress asymmetry) rather than cognitive processes or active decision-making akin to biological intelligence. As such, the device is best described as bio-inspired functionalities, rather than true “life-like” capabilities. The authors may consider justifying the performance of the soft actuator by comparing their soft robotics with *Gymnura micrura* intrinsically.

Second, in the undulation modes, only Region II (defect region) undergoes the undulation motion, the Region III still damps and oscillates driven by the buckling of the Region II. Is there any way to induce undulation of Region III? Can you reduce the bending rigidity of Region III to make it more flexible so undulation motion can translate from Region II to III?

Third, the adaptive locomotion and swimming of the soft robot across different terrains rely on two parallel lines to guide the motion as shown in Fig.4 E-G. In the reviewer’s opinion, such pre-design track limits the significance of its robotic functions. By strictly following fixed lines, the robot’s ability to autonomously adapt, make decisions, and exhibit adaptive intelligence is limited. This pre-defined path may mask the system’s true capability to navigate unpredictable environments, thus reducing the overall impact and novelty of the robot’s purported adaptiveness. The reviewer suggests redesigning the structure and setup of the soft robot based on the light-responsive soft actuator to make it freely move in different terrains.

Fourth, although in Fig. 5, the authors describe remote triggering, alternative materials, and extended feasibility for the actuator design, it seems peripheral to the core premise of creating an “environment adaptive multimodal actuator for life-like intelligent robotics.” The mention of broad-spectrum absorption, and erasable inks, does not convincingly establish the system’s immediate significance for intelligent robotic applications. In addition, in-depth investigation should be done for the dynamic application to show how the asymmetric actuator can benefit fluid transportation in detail. For the TENG application, the authors should show how to accurately control the voltage by tuning the environmental viscosity or temperature.

Reviewer #2

(Remarks to the Author)

This manuscript gives a detailed description of a photothermally driven multi-modal actuator. The tri-layer cantilever type actuator exhibits a well-known flapping mode oscillation under illumination mainly from a NIR laser. The tri-layer features an intentionally designed local defect in which one of the active outer layers is not bonded to the inner layer. This defect leads to the second mode of operation which involves buckling of the two bonded layers within this defect region, which then leads to a undulatory motion. Similar undulation motions have been observed in certain light responsive systems. The key contribution of this work is the clever design that combines these two modes into one composite actuator. The authors provide an excellent overview of the operation of this actuator. An exciting feature is the transitions between the different operation modes (a third mode is a simple light tracking mode) and the ability of this actuator to adapt to environmental cues to select different modes of operation. I am unaware of a work that combines operation modes in light responsive actuator in a similar manor. This work has the potential to inspire additional advances in multimodal actuators through combining designs involving known modes of operations, defect design, etc. Overall the data supplied in the main figures, extended figures, SI, and videos effectively supports the claims of the authors and meets the expected standards of the field for an experimental study.

A few comments related potential areas for improvement follow:

It would be helpful to give the main light source used in line 115 when constant illumination is listed (808 nm or NIR)

Lines 118-123 could be written a little more clearly. The section “was focused and eventually be transferred” on 119 was confusing (tense change?). The authors should make it clear the one third stress generated (line 121) is inferred from the DMA data and not from direct measurement or simulation.

It is not clear how turning the actuator upside down implies extension to other designs (line 143), possible the authors could be more specific.

Did delamination of the layers occur for d/L out of the 0.1-0.12 range? Line 198 seems to imply it and that should be reported if it's the case.

The use of buckling and undulation in the paper is sometimes confusing. For example, in line 202, the distinction between the tracking-undulation bifurcation and buckling performance was not clear to me.

Relating to the analytical model:

- 1) It would be helpful to the interested reader to have a free body diagram as part of the derivation of equations S1-S5 in section S2.1
- 2) It seems like the ratio K_0 should have α_1 and α_2 in it. They show up in equation (2) on line 219 of the main text, but also disappear there on the right side of the equation.
- 3) Overall a little more detail/clarity surrounding the derivation of K_0 and why it is the key bifurcation parameter (or the logic that led to that conclusion) would be helpful.

Briefly stating that the cart is suspended on wires around line 254 would be helpful.

It was not immediately clear what the “bow-shaped structure” was refereeing to in line 277.

In the Figure 2 caption (line 433-434) are the pink and purple areas the temperature during illumination? If so, please include that information in the caption for (B).

Figure 2C caption – it would be helpful to clarify how undulation frequency is defined. Or this could be placed in the main text.

Figure 2E caption – is amplitude measured using the point on the tip of the cantilever? It would be good to state this explicitly.

Figure 3H – it was not clear what the difference was between the two plots in H. I think it is light intensity. Either way including a label on the figure that makes the distinction clear would be helpful.

In section S1.2 the spacer for the LCE/CSNP appears to be 280 μm , but later I believe the layer is reported as 140 μm .

Figure S3 (B) – the caption seems to have a typo or grammar error that needs to be fixed.

Figure S15 – What do the different markers signify. Is the second X/L supposed to equal 0.1?

Fig S19 – A and B labels should be included in the caption so it is clearer which images are being discussed. Is the referenced video S9 or S10. Please check to make sure.

Reviewer #3

(Remarks to the Author)

This work presents a light-responsive actuator that can switch its motion modes between tracking, undulation, and oscillation. The actuator is made by introducing an unbound region in an elastomeric trilayer structure, and the motion mode is determined by the energy input. Theoretical modeling, simulation, and experiments are conducted to investigate the actuator's behavior. Mobile robots are built which can switch their locomotion modes in varying environments. This work is interesting but still needs to address several minor issues.

The following are some detailed comments:

1. In the Abstract and Introduction, the statement "current designs are constrained to single-mode actuation" isn't rigorous; please refer to the related works about multimodal actuators or robots.
2. In Fig. 1B, what's the difference between the solid line and the dashed lines? For the undulation branch (similarly for the tracking branch), can a state exist that has a larger energy beyond the undulation-oscillation bifurcation point but still performs the undulation?
3. In Fig. 2B, do the pink and purple areas refer to the responses under the same thermal energy input? In line 120, the statement of "sufficient" seems unclear for the input condition.
4. In Fig. S6B, the immediate temperature drop is explained to be caused by the energy release in structural buckling. Are there any references showing a similar buckling-induced temperature drop, or is it caused by the side effects of fast motion or self-shadowing? Please provide more support information.
5. In the caption of Fig. S15, double-check the values of the two groups of parameters.
6. When swimming in viscous fluid with a low Reynolds number, microorganisms use different but more efficient locomotion compared to the fish-like oscillation. In terms of speed or efficiency, can the swimming robot benefit from the automatic switch of motion modes? Please clarify the motivation for mode-switching.
7. In Fig. S17D, the vertical coordinate may infer that the tracking or undulation can have a higher output energy or speed than oscillation, which is a bit confusing.
8. In Fig. 2C, the data in the horizontal coordinate axis are a little crowded and unclear.
9. In lines 215-216, it would be clearer to clarify that Fig. 3E is the numerically simulated result.
10. In line 279, it should be "PDMS" rather than "PMDS".

Reviewer #4

(Remarks to the Author)

Version 1:

Reviewer comments:

Reviewer #1

(Remarks to the Author)

The revised manuscript well addressed each of my previous concerns. The authors have removed the potentially misleading "life-like intelligent" terminology, retitled the work, and inserted a quantitative comparison with *Gymnura micrura*. They also conducted additional experiments—varying Region III stiffness and adding a second defect—to show why undulation cannot yet propagate the full length of the actuator. For terrain adaptability, they tested alternative cart designs, analyzed friction-related failure modes, the reviewer acknowledged their effort to try different approaches to eliminate the pre-designed track, and looked forward to their future exploration towards more adaptive soft actuators. In summary, the manuscript is now scientifically sound, well-structured, and ready for publication. In light of these improvements, I am pleased to recommend the acceptance of this manuscript for publication in Nature Communications.

Reviewer #2

(Remarks to the Author)

The concerns raised in the original review have been adequately addressed and I have no new issues with the manuscript. I appreciate the authors' detailed and thorough responses.

Reviewer #3

(Remarks to the Author)

The authors have addressed my previous comments, and the manuscript has been significantly improved.

Reviewer #4

(Remarks to the Author)

Point-By-Point Response to Reviewers' Comments

(*Black italic: Reviewer's remarks; Blue: Our response; Black marked in yellow: Our revision in manuscript*)

Reviewer #1

- *In this manuscript, Chen et al. developed a light-responsive soft actuator with intrinsic geometric asymmetry (debonding defects in trilayer composite). Three modes of motion are identified: namely tracking, undulation, and oscillation. Through systematic experimental studies, the authors explored the optimal geometric parameters to allow the soft actuator to alter between three different modes upon varying input light intensity from the laser beam. The authors also demonstrated that motion modes can be switched based on environmental variation such as the changing of viscosity, temperatures, and physical contact. The authors showed some robotic functions based on its environmental adaptivity such as the soft robot can pass through the high-temperature area and swim from a low-viscosity area to a high-viscosity area. Lastly, the authors demonstrated the multifunction of such light-responsive soft actuator based on different motion modes in altering the airflow patterns, TENG. The light-responsive soft actuator can also work under different spectrums of laser input and the defect region can be replaced with more commercially available materials.*
- *The manuscript is well written and details of the design of the soft actuator are given. Such mode switch of light-driven soft actuator is sound. The mechanism governing the bifurcation phenomenon is well explained. It is an inspiring work and can attract attention from experts working in the soft robotics. However, the reviewer has several major comments on this work:*

Response:

We appreciate your positive and encouraging feedback. Our aim is to integrate geometric asymmetry into stimuli-responsive materials to enhance their response performance beyond a single-motion mode and to expand their functionalities and application fields. Your insightful and constructive comments have indeed prompted us to investigate more deeply and have significantly improved the overall quality of the manuscript.

- *First, While the presented soft robotic actuator demonstrate self-regulating and adaptive behaviors—switching between undulatory and oscillatory modes in response to energy states—characterizing it as “life-like intelligent” can be somewhat misleading. Fundamentally, the system relies on built-in mechanical feedback loops facilitated by material properties passively (e.g., photothermal effects, stress asymmetry) rather than*

cognitive processes or active decision-making akin to biological intelligence. As such, the device is best described as bio-inspired functionalities, rather than true “life-like” capabilities. The authors may consider justifying the performance of the soft actuator by comparing their soft robotics with *Gymnura micrura* intrinsically.

Response:

Thank you for your important insight regarding the concept of “life-like”. Recently, the terms “life-like” and “intelligence” have been increasingly used to describe biomimetic robotic behavior in stimuli-responsive materials-based systems¹⁻⁴. However, we agree that current designs, particularly those not reliant on algorithms, remain far from achieving bio-level intelligence. As you rightly pointed out, the use of “life-like” may be misleading, as our system does not exhibit cognitive processes or active decision-making. Therefore, to ensure precision and avoid misunderstanding, we have removed all instances of the term “life-like” and related descriptions from the Abstract, Introduction, and Results, and revised the title to “**Bio-inspired multimodal soft actuator with environmental self-adaptation**”.

Fig. R1: (A) Comparison of kinetic energy between undulations and oscillations. (B) Comparison of peak and average energy conversion efficiencies between undulation and oscillation modes.

On the other hand, to better compare our multimodal actuator with the intrinsic behavior of *Gymnura micrura*, we calculated both the peak and average output efficiencies of undulation (5.414 W/cm²) and oscillation (5.494 W/cm²) near the undulation-oscillation transition point based on modeling results (Fig. R1A). As shown in Fig. R1B, undulation exhibits a peak output more than twice that of oscillation, whereas oscillation yields a higher average output. This represents a trade-off similar to the behavioral strategy of *Gymnura micrura*, which uses undulatory swimming at lower speeds to enhance instantaneous output and maneuverability during predation or reproduction, and switches to oscillatory swimming for high-speed cruising⁵. Besides, the motion-switching ability enables our actuator to maintain continuous locomotion after encountering physical contact (Fig. 4E, Fig. S23, and Movie S11), which is

also similar to the behavior of *Gymnura micrura*, where low-speed undulatory swimming is preferred in confined spaces to improve maneuverability and control. Thus, the relevant descriptions in the Introduction and Results sections have been revised as follows: “The availability of motion switching creates a dynamic photo-thermo-mechanical feedback system with the potential for multimodal adaptation under varying conditions (e.g., light intensity, fluid viscosity, surrounding temperature, and physical constraint) controlled by the energy state. This leads to distinct performance outputs—analogueous to the behavioral strategy of *Gymnura micrura*—with higher peak output during undulation but higher average output during oscillation, allowing for continuous locomotion across diverse environments”, “Furthermore, we observe bursts of energy output every cycle in undulation mode, while a higher average kinetic energy output is presented in oscillation despite lower energy maxima near the undulation–oscillation bifurcation point (Fig. 3F and Supplementary Text 2.3). Specifically, the peak output efficiency of undulation is more than twice that of oscillation, while its average output efficiency is only 0.681 times that of oscillation (Supplementary Fig. 19)—a trade-off akin to the behavioral strategy of *Gymnura micrura*, which employs undulatory swimming at lower speeds to enhance instantaneous output and maneuverability during predation or reproduction, and shifts to oscillatory swimming under high-speed cruising⁵” and “This undulation of the asymmetric actuator indicated its ability to achieve the higher maneuverability to navigate obstacles under lower energy output, paralleling the undulatory locomotion of *Gymnura micrura* that adopts this strategy at low speeds to improve controllability in confined spaces.”

- *Second, in the undulation modes, only Region II (defect region) undergoes the undulation motion, the Region III still damps and oscillates driven by the buckling of the Region II. Is there any way to induce undulation of Region III? Can you reduce the bending rigidity of Region III to make it more flexible so undulation motion can translate from Region II to III?*

Response:

Thank you for your constructive comment. To answer your question, we have explored two different approaches to increase the undulation tendency of Region III: (i) changing the length of Region III while keeping the lengths of Region I (X) and Region II (d) constant to reduce the bending rigidity/lateral stiffness, and (ii) introducing a second defect within Region III.

Table R1. Motion comparison of different asymmetric trilayers with varying lengths of Region III.

Length of the Region III (mm)	12	17 (original)	22	27	32	37
Initial state						
Lateral stiffness (MPa)	39.3	15.1	7.8	4.2	2.5	1.5
Tracking	√	√	√	√	√	×
Undulation	×	√	√	√	√	×
Oscillation	×	√	√	√	×	×

First, under a uniformly distributed load G , the maximum deflection δ_{\max} at the free end of a cantilever beam of length L is given by

$$\delta_{\max} = \frac{GL^4}{8EI} = \frac{3GL^4}{2Ebh^3} \quad (\text{R1})$$

where E is Young's modulus of the beam and I is the moment of inertia; for a rectangular cross-section of width b and thickness of h , $I = bh^3/12$. The beam's lateral stiffness is defined as the ratio of the applied load G to the δ_{\max} :

$$k_{\text{lateral}} = \frac{G}{\delta_{\max}} = \frac{G}{3GL^4 / 2Ebh^3} = \frac{2Ebh^3}{3L^4} \quad (\text{R2})$$

For the first groups, as shown in Table R1, with the increasing length of Region III while keeping the length (d) and location (X) of the defect constant, the lateral stiffness of the asymmetric trilayer decreases step by step, as calculated by Eqn. (R2). As shown in Table R1, four distinct performance outcomes were experimentally observed based on the length of Region III: (i) With a short length (12 mm) and high stiffness, the cantilever exhibits only static tracking. (ii) With moderate stiffness, it can achieve all three motion modes and switch between them according to light intensity, as demonstrated in the cases we showed in the manuscript. (iii) When the stiffness is too low (as in the case with a Region III length of 32 mm), the Region III becomes too soft to bend upward and block the light, due to both reduced stiffness and the increased gravitational effect caused by a right-shifted center of mass. In this case, the feedback loop cannot be completed, preventing self-sustained motion, let alone the occurrence of bifurcation. (iv) With a further increase in length (37 mm), even tracking becomes unachievable, as the synergistic effect of increased gravity and reduced rigidity makes it difficult for the cantilever to support itself.

Fig. R2: Multimodal behavior of the asymmetric trilayer with two defect regions. (A) An additional defect (Region IV) was introduced within Region III. (B) The presence of the second defect affected all three motion modes and introduced an additional oscillation frequency in the oscillation mode.

On the other hand, we also explored the design with a second defect introduced in Region III (located 10 mm from the fixed end and 1.5 mm in width, as shown in Fig. R2A) to increase the tendency of buckling and undulation. As shown in Fig. R2B and Movie S15, although the asymmetric trilayer was able to exhibit three distinct motion modes, it was still not able to transfer the undulation into Region III. Theoretically, the absence of the undulation mode of Region III containing a second defect (Region IV) can be attributed to two factors. First, the laser's limited spot size prevented sufficient irradiation of region III, so there was not enough thermal strain to drive deformation at that defect. Second, the structural connection between the first defect in Region II and the second defect in Region IV impeded the propagation of the traveling wave generated from Region II into Region III.

Meanwhile, the added defect introduced a new oscillation frequency in the oscillation mode, indicating an additional dynamic response within the system, which we believe is worth discussing. Therefore, we have included it in the Results section, along with Supplementary Fig. 33 and Movie S15, as follows: “Moreover, we attempted to introduce additional defects into Region III and found that the presence of a second defect could influence all three motion modes and induce an additional oscillation frequency in the oscillation mode (Supplementary Fig. 33 and Movie S15)” and “**Movie S15. The motion adaptation of the asymmetric trilayer with a second defect.** The asymmetric trilayer with two defects was fixed on the stage, and the NIR light was irradiated from the right side horizontally, with the light intensity increased from 2.925 to 3.728 W/cm² step by step. All motions were played in a real-time manner.”

Based on the above experimental and modeling results from modification with current structure, undulation still only occurs in Region II and cannot propagate into Region III in a highly reliable manner yet. However, your suggestion has provided valuable inspiration. As a next step, we plan to explore other structural designs that could achieve both full-body undulations and multimodal adaptation.

- *Third, the adaptive locomotion and swimming of the soft robot across different terrains rely on two parallel lines to guide the motion as shown in Fig.4 E-G. In the reviewer's opinion, such pre-design track limits the significance of its robotic functions. By strictly following fixed lines, the robot's ability to autonomously adapt, make decisions, and exhibit adaptive intelligence is limited. This pre-defined path may mask the system's true capability to navigate unpredictable environments, thus reducing the overall impact and novelty of the robot's purported adaptiveness. The reviewer suggests redesigning the structure and setup of the soft robot based on the light-responsive soft actuator to make it freely move in different terrains.*

Response:

Fig. R3: (A) Two revised cart designs for terrestrial locomotion attempts. (B) Air-bubble-enabled underwater design achieving horizontal locomotion under overhead light irradiation. (C) Spherical design capable of freely rolling across various terrains.

Thanks for your valuable suggestion. For terrestrial locomotion across different terrains, following your recommendation, we redesigned two modified carts based on our previous design: one with a PET wing and one without (Fig. R3A). In the non-wing design, periodic contact with the glass slide failed to generate sufficient force to propel the cart forward—even when the light intensity was increased to 6.939 W/cm^2 until the LCE was damaged—primarily due to the high friction between the feet and the glass slide (Movie R1, left). In contrast, the suspended design (Fig. 4E), which included a PET wing and was guided by copper wires, required only minimal vibrations between the perforated cart frame and the copper wires under a lower light intensity of 4.772 W/cm^2 , effectively converting vibration into forward motion by overcoming the reduced friction. On the other hand, although vibrations were observed in the cart with a PET wing before the sample burned (Movie R1, right), contact between the wing and the glass slide dissipated most of the energy, leaving the cart immobilized due to friction.

Moreover, as you pointed out, we agree that the precise alignment of light incidence and the reliance on a pre-defined path are indeed critical limitations for light-responsive soft actuators based on the self-shadowing mechanism. This theoretically is a fundamental limitation of all

cantilever structures relying on the self-shadowing mechanism, as they must follow strictly predefined paths due to the requirement for accurately aligned light irradiation⁶⁻⁹. To address the limitations of such structures, we are actually exploring new strategies beyond this cantilever structure: (i) we are trying to apply surface coatings to trap air bubbles underwater, using total internal reflection at the air-water interface to decouple the locomotion direction from the light path, *i.e.*, the underwater walker can freely move horizontally fueled by the overhead vertical irradiation (Fig. R3B). (ii) Meanwhile, we are designing a spherical soft robot that is capable of freely rolling across various terrains, enabling omnidirectional movement beyond the pre-defined paths (Fig. R3C).

- *Fourth, although in Fig. 5, the authors describe remote triggering, alternative materials, and extended feasibility for the actuator design, it seems peripheral to the core premise of creating an “environment adaptive multimodal actuator for life-like intelligent robotics.” The mention of broad-spectrum absorption, and erasable inks, does not convincingly establish the system’s immediate significance for intelligent robotic applications. In addition, in-depth investigation should be done for the dynamic application to show how the asymmetric actuator can benefit fluid transportation in detail.*

Response:

Thank you for your valuable comment. For a more in-depth investigation, we conducted additional experiments in which the asymmetric actuator was placed in a fluid environment containing small particles. The PIV results have demonstrated undulation increases the fluid velocity above the defect area, while oscillation generates a vortex near the tip region, which confirms distinct interactions between the actuator and the surrounding fluid (Fig. R4A-B). Furthermore, the asymmetric actuator can interact differently with suspended particles—oscillation-induced vortices can trap small particles, whereas undulation primarily drives upward thermal flow, causing particles to rise (Fig. R4C-D and Movie R2). These findings further demonstrate the asymmetric actuator’s ability to influence fluid transport both with and without particles. As a result, Fig. R4C-D has been integrated into Fig. 5B, and the corresponding discussion has been revised as follows: “This is analogous to the non-reciprocal vortex existing in oscillating fishtails that generates thrust, which highlights the potential of our asymmetric actuator for fluid dynamics applications, including fluidic transportation and particle trapping during oscillation in fluid environments (Fig. 5B)^{10,11}.”

Fig. R4: PIV results showing the distinct interaction generated by the asymmetric actuator during (A) undulation and (B) oscillation. Different interactions between the asymmetric actuator and particles during (C) undulation and (D) oscillation.

- *For the TENG application, the authors should show how to accurately control the voltage by tuning the environmental viscosity or temperature.*

Response:

For the TENG demonstration, following your insightful suggestion, we first tried to control the voltage output by tuning the environmental temperature. As shown in Fig. R5A, the entire setup was placed in a high-temperature oven, with a thermocouple positioned near the TENG to accurately monitor the temperature. As the temperature increased, the undulation amplitude also increased, resulting in a larger air gap during buckling and consequently higher open-circuit voltage (Fig. R5B). However, when the temperature reached 46.2 °C, the asymmetric trilayer switched to oscillation mode, where air gap changes became nearly undetectable, leading to negligible output. Specifically, both the average and peak output voltages, calculated using the following equations, increased step by step with temperature in the undulation mode. This indicates that for a given light intensity, the TENG output can be effectively modulated by temperature, and when the temperature exceeds a critical threshold, it triggers the undulation-oscillation bifurcation, causing the output to abruptly drop to zero and function as a temperature-controlled switch.

$$V_{\text{peak}} = \max |V(t)|$$

$$V_{\text{avg}} = \frac{1}{T} \int_{t_0}^{t_0+T} V(t) dt$$

Therefore, the relevant discussion has been revised in the Manuscript as follows: “This distinct bifurcation can be applied not only to monitor specific environmental changes—such as voltage drops indicating viscosity or temperature crossing the threshold—but also to control desired voltage outputs (both average and peak voltage output) by manually tuning environmental conditions (Fig. 5E).”

Fig. R5: (A) Experimental setup for TENG performance testing under different temperatures. (B) Output voltage of the TENG under different temperatures. (C) Comparison of average and peak output voltages of the TENG under different temperatures.

Meanwhile, we also attempted to tune the output voltage by placing the setup in an aquatic environment to vary viscosity, but no effective signal was detected, mainly because the charge could not accumulate in the liquid medium. In addition, we tried encapsulating the multilayered TENG to prevent charge leakage; however, the added encapsulation increased structure rigidity largely, restricting the dynamic deformation for the contact-separation mechanism.

Fig. R6: Revised version of Figure 5 in Manuscript.

As a brief summary, the in-depth investigations on particle interaction and temperature-controlled output were successfully conducted and integrated into the revised Fig. 5E (Fig. R6). Additionally, since our goal is to propose a generalized strategy for multimodal, switchable designs, the exploration of generalizability using alternative irradiation sources, material systems, and erasable inks remains relevant and worth discussing; therefore, we kept Fig. 5F-5G as the final part of the Results section. Meanwhile, the relevant section in the Supplementary Materials has been revised as follows: “The open-circuit voltage of the photothermal TENGs was measured using the open-circuit potential-time mode of the electrochemical workstation. The entire setup was placed in a high-temperature oven equipped with a thermocouple to evaluate the output voltage under different temperature conditions. The peak voltage (V_{peak}) and average voltage (V_{avg}) were calculated using the following equations:

$$V_{\text{peak}} = \max |V(t)|$$

$$V_{\text{avg}} = \frac{1}{T} \int_{t_0}^{t_0+T} V(t) dt$$

Reference

- 1 Wang, X.-Q. & Ho, G. W. Design of untethered soft material micromachine for life-like locomotion. *Materials Today* **53**, 197-216 (2022).
- 2 Zhao, Y. *et al.* Twisting for soft intelligent autonomous robot in unstructured environments. *Proceedings of the National Academy of Sciences* **119**, e2200265119 (2022).
- 3 Sitti, M. Physical intelligence as a new paradigm. *Extreme Mechanics Letters* **46**, 101340 (2021).
- 4 Li, M., Pal, A., Aghakhani, A., Pena-Francesch, A. & Sitti, M. Soft actuators for real-world applications. *Nature Reviews Materials* **7**, 235-249 (2022).
- 5 Rosenberger, L. J. Pectoral Fin Locomotion in Batoid Fishes: Undulation Versus Oscillation. *Journal of Experimental Biology* **204**, 379-394 (2001).
- 6 Zhao, Y. *et al.* Sunlight-powered self-excited oscillators for sustainable autonomous soft robotics. *Science Robotics* **8**, eadf4753 (2023).
- 7 Gelebart, A. H. *et al.* Making waves in a photoactive polymer film. *Nature* **546**, 632-636 (2017).
- 8 Zhao, Y. *et al.* Soft phototactic swimmer based on self-sustained hydrogel oscillator. *Science Robotics* **4**, eaax7112 (2019).
- 9 Cheng, M. *et al.* Light-Fueled Polymer Film Capable of Directional Crawling, Friction-Controlled Climbing, and Self-Sustained Motion on a Human Hair. *Advanced Science* **9**, 2103090 (2022).
- 10 Deng, Z., Zhang, H., Priimagi, A. & Zeng, H. Light-Fueled Nonreciprocal Self-Oscillators for Fluidic Transportation and Coupling. *Advanced Materials* **36**, 2209683 (2022).
- 11 Sfakiotakis, M., Lane, D. M. & Davies, J. B. C. Review of fish swimming modes for aquatic locomotion. *IEEE Journal of Oceanic Engineering* **24**, 237-252 (1999).

Reviewer #2

- *This manuscript gives a detailed description of a photothermally driven multi-modal actuator. The tri-layer cantilever type actuator exhibits a well-known flapping mode oscillation under illumination mainly from a NIR laser. The tri-layer features an intentionally designed local defect in which one of the active outer layers is not bonded to the inner layer. This defect leads to the second mode of operation which involves buckling of the two bonded layers within this defect region, which then leads to a undulatory motion. Similar undulation motions have been observed in certain light responsive systems. The key contribution of this work is the clever design that combines these two modes into one composite actuator. The authors provide an excellent overview of the operation of this actuator. An exciting feature is the transitions between the different operation modes (a third mode is a simple light tracking mode) and the ability of this actuator to adapt to environmental cues to select different modes of operation. I am unaware of a work that combines operation modes in light responsive actuator in a similar manor. This work has the potential to inspire additional advances in multimodal actuators through combining designs involving known modes of operations, defect design, etc. Overall the data supplied in the main figures, extended figures, SI, and videos effectively supports the claims of the authors and meets the expected standards of the field for an experimental study.*

Response:

Thank you for your thoughtful and encouraging comments, which will certainly help improve the quality of the manuscript. We truly appreciate your understanding of our multimodal design with environmental adaptation, and we are pleased that the data organization and supporting materials met your expectations.

- *A few comments related potential areas for improvement follow:*
- *It would be helpful to give the main light source used in line 115 when constant illumination is listed (808 nm or NIR)*

Response:

Thanks for your detailed comments. We have revised the relevant paragraph as follows: “When the trilayer was first exposed to constant **near-infrared (NIR)** illumination from the unconstrained side, the top bilayer reached a higher temperature (69.7-82.3 °C) compared to the bottom LCE/CSNP monolayer (69.1-78.7 °C) because the tip of the trilayer naturally dipped downward due to gravity.”

- *Lines 118-123 could be written a little more clearly. The section “was focused and eventually be transferred” on 119 was confusing (tense change?). The authors should make it clear the one third stress generated (line 121) is inferred from the DMA data and not from direct measurement or simulation.*

Response:

Thank you for pointing out our tense error and the lack of clarification that the curves in Fig. 2B were derived from DMA data, and we apologize for any misunderstanding this may have caused. We have revised the relevant description as follows: “In Region II, despite sufficient thermal energy and a higher temperature in the bilayer, it generated less than one-third of the stress (77.1-134.9 kPa) compared to the LCE/CSNP monolayer (362.9-475.6 kPa), as indicated by the DMA results (Fig. 2B).”

- *It is not clear how turning the actuator upside down implies extension to other designs (line 143), possible the authors could be more specific.*

Response:

Thank you for the constructive suggestion. As shown in Fig. S11, the reversed structure—with the defect in the upper half—still exhibited similar undulatory behavior, with the wave propagating from the irradiation side and the tip bending upward against gravity. These observations confirm that gravity does not play a dominant role in driving the undulation. Rather, the key factors are the structural asymmetry and the photo-thermo-mechanical coupling. This finding can simplify the design of complex systems beyond cantilever beams, such as three-dimensional architectures, and extend their potential applications to low- or zero-gravity environments.

To improve clarity and maintain logical flow, we have relocated this discussion on structural extension to the Discussion section. The paragraph has been revised as follows: “Afterward, we turned the trilayer upside down and observed the similar continuous undulation, with the wave propagating along the structure from the unconstrained side, while the tip was even able to buckle against the direction of gravity (Supplementary Fig. 11). This suggests that gravity is not the dominant factor in this instability-induced behavior; instead, the primary contributors are the structural asymmetry and the photo-thermo-mechanical coupling.”

- *Did delamination of the layers occur for d/L out of the 0.1-0.12 range? Line 198 seems to imply it and that should be reported if it's the case.*

Response:

Fig. R7: (A) Comparison of adhesion performance among different commercial high-temperature silicone glues. (B) Influence of structural parameters on multimodal motion (red area indicates where undulation occurs but without oscillation). (C) Comparison of numerically simulated K and K0 (white region refers to where the undulation can occur)

Thank you for the detailed question. The answer is yes. Specifically, for the groups with $X/L = 0.20$, three distinct cases were observed: (i) For $d/L < 0.10$, the temperature and stress differences were insufficient to trigger the bifurcation, resulting in only static tracking; (ii) for $0.10 < d/L < 0.12$, the stress difference was sufficient to initiate the bifurcation, and the adhesion between the trilayers in Regions I and III was adequate to maintain structural integrity; (iii) for $d/L > 0.12$, although the stress difference was high enough, the buckling energy exceeded the adhesive limit, leading to delamination of the trilayer. We admit that, despite our efforts to enhance adhesion—including the use of various high-temperature silicon glues—delamination still occurred even with the optimal adhesive (Permatex 82194 in Fig. R7A). Similar delamination phenomena were observed in the red area of Fig. R7B with increased d/L , attributed to insufficient adhesion. If the adhesive strength had been adequate, these groups would likely exhibit bifurcation, as simulated by our modeling (Fig. 3E and Fig. R7C). The relevant discussion has been revised as follows: “However, only the range $0.10 < d/L < 0.12$ successfully achieved the undulation-oscillation bifurcation while delamination occurred in trilayers with $d/L > 0.12$ due to excessive energy exceeding the adhesive limit and destroying structural integrity.”

- *The use of buckling and undulation in the paper is sometimes confusing. For example, in line 202, the distinction between the tracking-undulation bifurcation and buckling performance was not clear to me.*

Response:

Thank you for the comment. For undulation, Buckling is a necessary condition for the generation of undulation (Fig. 2A). Without buckling, the trilayer can only exhibit static tracking or reciprocal oscillation. Moreover, the bifurcation arises from the competition between buckling and bending tendencies, which is governed by structural asymmetry (Eqn. 1). Specifically, only when the buckling tendency dominates, the system can undergo buckling,

leading to self-sustained undulation; otherwise, it results in tracking or oscillation. Therefore, for samples with defect parameters in the regime that does not permit buckling, the undulation state cannot occur, let alone a tracking-undulation or undulation-oscillation bifurcation. In the sentence you mentioned, we try to emphasize that for a fixed d/L , the location of the defect X/L determines whether a tracking-undulation bifurcation occurs, rather than merely affecting the characteristics of undulation such as amplitude or frequency. We admit that the original phrase “buckling performance” was not precise, and we have revised the corresponding paragraph as follows: “On the other hand, when d/L was fixed at 0.12, bifurcation occurred only within the range $0.20 < X/L < 0.30$, with the resulting undulations exhibiting similar amplitudes and frequencies (Fig. 3C and Supplementary Fig. 17B). This indicated that the location of the defect region primarily determined whether a tracking-undulation bifurcation occurred, rather than simply influencing the undulation characteristics such as amplitude or frequency.”

- *Relating to the analytical model:*

1) *It would be helpful to the interested reader to have a free body diagram as part of the derivation of equations S1-S5 in section S2.1*

Response:

Fig. R8: Free body diagram of the composite structure with an artificial defect.

Thanks for this valuable comment. According to your suggestion, we have added a free body diagram to clarify the derivation of our theoretical bifurcation criterion as Fig. S1 in Supplementary Text 2.1 (Fig. R8). In Supplementary Text 2.1, Eqs. S1 and S2 describe the balance of forces P_i and moments M_i in the structure, respectively. Eq. S3 then relates the bending moment to the radius of curvature ρ . Eqs. S4 and S5 enforce strain-compatibility: Eq.

S4 requires identical strain the the interface between layers 1 and 2, and Eq. S5 requires identical strain throughout the bonded regions between layers 2 and 3.

2) *It seems like the ratio K_0 should have α_1 and α_2 in it. They show up in equation (2) on line 219 of the main text, but also disappear there on the right side of the equation.*

Response:

Thanks for your careful review and allow us to clarify Eq. (2), as shown below:

$$K_0 = \frac{\alpha_3 \Delta T_{23}}{\alpha_1 \Delta T_{12}} = \frac{\alpha_{\text{LCE/CSNP}} (T_{23} - T_0)}{\alpha_{\text{LCE/CSNP}} (T_{12} - T_0)}, \quad \begin{cases} K > K_0, \text{ undulation} \\ K < K_0, \text{ oscillation} \end{cases}$$

In this equation, α_1 and α_3 denote the coefficients of thermal expansion (CTE) of layers 1 and 3, respectively. Because both layers are fabricated from the same LCE/CSNP recipe under identical synthesis conditions, they have the same CTE (*i.e.*, $\alpha_1 = \alpha_3$), allowing these terms to cancel on the right-hand side. If, as you note, layers 1 and 3 have different CTE, the α_i terms cannot be eliminated. To make this explicit, we have revised the manuscript as follows: “**Since the top and bottom layers are fabricated from the same LCE/CSNP recipe under identical synthesis conditions (*i.e.*, $\alpha_1 = \alpha_3 = \alpha_{\text{LCE/CSNP}}$), these terms cancel out in Eqn. (2).** Thus, we can obtain a K_0 map for given geometrical parameters, **Fig. 3D** (see details in Supplementary Text 2.2).”

3) *Overall a little more detail/clarity surrounding the derivation of K_0 and why it is the key bifurcation parameter (or the logic that led to that conclusion) would be helpful.*

Response:

We appreciate the opportunity to elaborate on the derivation and physical meaning of the bifurcation parameter K_0 . We define K_0 as the ratio of the thermal strain in layer 3 (the lower LCE/CSNP layer) at the layer 2-3 interface ($\alpha_3 \Delta T_{23}$) to the thermal strain in layer 1 (the upper LCE/CSNP layer) at the layer 1-2 interface ($\alpha_1 \Delta T_{12}$). Experimentally, we found that all actuators in the oscillation mode directly bend upward after a few seconds of irradiation, whereas those in the undulation mode initially bend downward—driven by greater contraction and buckling in the unbonded region of the layer 3—before subsequently bending upward. The contrasting bending directions indicate that thermal strain ratio K_0 is likely to serve as a bifurcation parameter: when the contraction at the 1-2 interface exceeds that at the 2-3 interface, the actuator transitions directly to oscillation; otherwise, the actuator first bends downward and then upward, leading to undulation. We derive the expressions for the thermal strain ratio K_0 from force and moment balances (Eqs. S1-S2), the moment-curvature relationship (Eq. S3),

and the assumption of perfect bonding at the 1-2 interface and in the bonded regions of the 2-3 interface (Eqs. S4-S5). In the revised manuscript and Supplementary Materials, we have provided more details of the derivation and justified the selection of the bifurcation parameter by adding a free-body diagram and defining each variable explicitly, as follows: “The critical ratio K_0 between ΔT_{23} and ΔT_{12} [Eq. (S6)] delineates the transition between bending (oscillation) and buckling (undulation), which can be obtained by solving the equilibrium and strain compatibility conditions embodied in Eqns. **Error! Reference source not found.-Error! Reference source not found.**”

$$K_0 = \frac{\Delta T_{23}^{\text{analy}}}{\Delta T_{12}^{\text{analy}}} = \frac{a_1 E_1 (a_1 a_2 E_2 L_0 + a_2^2 E_2 L_0 - a_1 a_2 E_2 d_0 - a_2^2 E_2 d_0 + a_1 a_3 E_3 L_0 + 2 a_2 a_3 E_3 L_0 + a_3^2 E_3 L_0)}{a_3 E_3 L_0 (a_1^2 E_1 + 2 a_1 a_2 E_1 + a_1 a_3 E_1 + a_2^2 E_2 + a_2 a_3 E_2)} \quad (\text{S3})$$

Experiments can also provide a ratio K with the same definition $\Delta T_{23}^{\text{exp}}/\Delta T_{12}^{\text{exp}}$, while the interfacial temperature changes, ΔT_{12} and ΔT_{23} , can be measured directly using an IR camera. When the K is greater than the theoretical ratio K_0 , which suggests that layer 3 contracts more than layer 2, the whole structure bends down and undulates. Conversely, if $K < K_0$, the structure bends up and oscillates.

Due to the limited resolution of the IR camera, it is difficult to obtain the exact temperature at the interfaces. Therefore, thermomechanical numerical simulations based on COMSOL Multiphysics were performed to observe the temperature changes during motions and obtain $\Delta T_{12}^{\text{exp}}$ and $\Delta T_{23}^{\text{exp}}$. Strains and stresses were simulated by Solid Mechanics module, and the temperature distributions were modeled in Heat Transfer in Solids interface. A multiphysics interface, Thermal Expansion, was employed to apply thermal strain to the structure, which is the link between thermal variable T and mechanical variable ϵ_{th} .”

- Briefly stating that the cart is suspended on wires around line 254 would be helpful.

Response:

Thank you for the detailed comments. We have added the relevant description to the manuscript, in addition to Supplementary Text 1.9, to clarify this point, as shown below: “To demonstrate its application, we first integrated the asymmetric trilayer as an actuator with a PET film into a cart frame made of Balsa wood, which was perforated and suspended on the copper wires to guide the locomotion direction (Fig. 4E).”

- It was not immediately clear what the “bow-shaped structure” was referring to in line 277.

Response:

Thank you for the detailed comments. The bow-shaped structure refers to the two-end fixed structure for undulation, as mentioned in Introduction (Supplementary Fig. 4C) and the section Tracking-undulation bifurcation (Supplementary Fig. 9). To improve clarity, the relevant description has been revised as follows: “Additionally, the periodic vacancy changes in the bow-shaped structure **during buckling-induced undulation** make the asymmetric actuator promising for developing triboelectric nanogenerators (TENG) by adopting PTFE/Pt and PMDS/Pt as inner layers (Supplementary Fig. 28).”

- *In the Figure 2 caption (line 433-434) are the pink and purple areas the temperature during illumination? If so, please include that information in the caption for (B).*

Response:

For the question, the answer is yes. The pink and purple areas in Fig. 2B represent the temperature variation range during illumination, as detailed in Supplementary Fig. 8A. The revised caption is as follows: “**(B)** Actuation stress of LCE/CSNP monolayer and bilayer under DMA isostrain condition (pink and purple regions refer to the temperature range of LCE/CSNP monolayer and bilayer **during illumination**, respectively).”

- *Figure 2C caption – it would be helpful to clarify how undulation frequency is defined. Or this could be placed in the main text.*

Response:

Thank you for the detailed comment. The undulation frequency is defined as the frequency of each cycle beginning with buckling and followed by wave propagation before the onset of the next buckling. Thus, it is therefore equivalent to the buckling frequency. This definition has been added to the caption of Fig. 2C, as follows: “**(C)** Comparison of tip amplitude and frequency over a long time, **where the undulation frequency is defined as the buckling frequency.**”

- *Figure 2E caption – is amplitude measured using the point on the tip of the cantilever? It would be good to state this explicitly.*

Response:

For the question, the answer is yes—all trajectories were measured using the tip of the cantilever unless otherwise specified (like Fig. 2D). According to your suggestion, the caption has been revised as below: “**(E)** Amplitude adaptation to increasing input, **calculated from the trajectory of the cantilever tip.** (transparent region for undulation while pink region for oscillation. The maximum amplitude is the buckling amplitude in undulation or the oscillation

amplitude in oscillation; the residual amplitude is the amplitude of the last complete swing before buckling in undulation, which does not exist in oscillation).”

- *Figure 3H – it was not clear what the difference was between the two plots in H. I think it is light intensity. Either way including a label on the figure that makes the distinction clear would be helpful.*

Response:

Fig. R9: Revised version of Figure 3H in Manuscript.

Thank you for your comment. You are absolutely right regarding Fig. 3H—we aimed to demonstrate that the asymmetric trilayer can self-excite and autonomously adopt the optimal motion mode in response to different light intensities: 3.728 W/cm² for undulation and 5.735 W/cm² for oscillation, as shown in Supplementary Movie 5. As suggested, to better convey this concept and clarify the distinction between the two modes, we have added a label to Fig. 3H (Fig. R9).

- *In section S1.2 the spacer for the LCE/CSNP appears to be 280 μm, but later I believe the layer is reported as 140 μm.*

Response:

Sorry for the misunderstanding. As you correctly noted, the spacer used for the LCE/CSNP was 280 μm, resulting in an initial film thickness of 280 μm after the first-step thermal crosslinking. Since a uniaxial stretching of 100% strain was applied and maintained during the second-step UV curing, the film thickness was reduced by half to 140 μm—this is the final thickness of the LCE/CSNP monolayer used for assembly. To avoid further confusion, we have revised the relevant description as follows: “After 3 min of reaction, the mixture was cast into two tape-covered glass slides with a 280 μm spacer and cured at 60 °C overnight. The resulting LCE/CSNP film was then peeled from the mold and subjected to uniaxial stretching to 100% strain, followed by photocrosslinking under a UV lamp (UVGO, 365 nm) at an intensity of 35

mW/cm² for 2 hours on each side. As a result, a crosslinked LCE/CSNP monolayer with a final thickness of 140 μm was prepared for the subsequent assembly step.”

- *Figure S3 (B) – the caption seems to have a typo or grammar error that needs to be fixed.*

Response:

Thanks for your careful reading. We have checked the grammar and revised the caption of Fig. S3 (now Fig. S5) as follows: “**Fig. S5. Schematic illustration for the preparation of the asymmetric trilayer structure.** (A) The whole structure was based on a symmetrical active-passive-active trilayer structure with an artificially introduced defect by partially blocking the silicon glue (see Section 1.3). (B) The active layers were composed of LCE with candle soot nanoparticles (CSNPs) that were dispersed as the photothermal agent (see Section 1.2). Dynamic light scattering (DLS) analysis revealed an average CSNP size of approximately 8.31 nm, consistent with the TEM image.”

- *Figure S15 – What do the different markers signify. Is the second X/L supposed to equal 0.1?*

Response:

Thank you for your question. The red hollow symbols represent the state where $K < K_0$, indicating that the bending tendency dominates and the trilayer remains in the tracking or oscillation state. In contrast, the solid symbols correspond to $K > K_0$, where the buckling tendency dominates, enabling the undulation. Additionally, as you correctly pointed out, the second X/L in the original caption should be 0.10. To better convey the meaning of Fig. S15 (now Fig. S18), we have revised the caption as follows: “**Fig. S18. Comparison of the relationship between K and K_0 .** The red hollow symbols represent states where $K < K_0$, indicating that the bending tendency dominates and the trilayer remains in the tracking or oscillation state. In contrast, the solid symbols indicate $K > K_0$, where the buckling tendency dominates, enabling the undulation. For X/L = 0.20 and d/L = 0.12, K can exceed the critical K_0 (blue dotted line), leading to the undulation. In contrast, for groups where K consistently remains below K_0 (e.g., X/L = 0.10 and d/L = 0.12; purple dotted line), the systems are limited to exhibiting only tracking and oscillation.”

- *Fig S19 – A and B labels should be included in the caption so it is clearer which images are being discussed. Is the referenced video S9 or S10. Please check to make sure.*

Response:

Thank you for the comment. Fig. S19A (now Fig. S23A) illustrates that, following self-excitation—a capability not present in the symmetric trilayer—the asymmetric trilayer could achieve self-continuous locomotion under a lower energy input (3.967 W/cm^2), without being obstructed by the physical constraint, as confirmed by tracking the tip trajectory in Movie S10. In contrast, Fig. S19B (now Fig. S23B) shows that under the same conditions, even with a higher light intensity (6.732 W/cm^2), the cart based on the symmetric trilayer was blocked by the constraint, with most of the input energy dissipated through repeated contacts against the barrier as shown in the comparison video (Movie S11). To make this clearer, we have revised the caption following your suggestion: “**Fig. S23. Comparison of (A) asymmetric and (B) symmetric trilayer for cart demonstrations.** (A) The asymmetric trilayer, following self-excitation—a capability absent in the symmetric trilayer—achieved self-continuous locomotion under a lower energy input (3.967 W/cm^2), without being obstructed by the physical constraint (Supplementary Movie S10). (B) In contrast, under the same conditions, even with a higher light intensity (6.732 W/cm^2), the cart equipped with the symmetric trilayer was blocked by the constraint, with most of the input energy dissipated through repeated contact with the constraint (Supplementary Movie S11).”

Reviewer #3

- *This work presents a light-responsive actuator that can switch its motion modes between tracking, undulation, and oscillation. The actuator is made by introducing an unbound region in an elastomeric trilayer structure, and the motion mode is determined by the energy input. Theoretical modeling, simulation, and experiments are conducted to investigate the actuator's behavior. Mobile robots are built which can switch their locomotion modes in varying environments. This work is interesting but still needs to address several minor issues.*

Response:

Thanks for your insightful feedback and interest in our work. As you noted, our aim is to demonstrate a simple yet generalizable design for achieving a multimodal actuator with self-regulated motion-switching capability. We appreciate your detailed comments and questions, which will definitely help us further refine the manuscript.

- *The following are some detailed comments:*
 1. *In the Abstract and Introduction, the statement “current designs are constrained to single-mode actuation” isn't rigorous; please refer to the related works about multimodal actuators or robots.*

Response:

Thanks for your constructive comment. We apologize for the ambiguity in the term “current designs”—it specifically refers to the previously mentioned *autonomous soft robots with built-in feedback control* that are fully constructed from soft materials, without the integration of rigid components or algorithm-based control, and are triggered solely by constant external stimuli.

We acknowledge that with the aid of chips and AI algorithms, conventional autonomous robots—such as the quadruped Spot¹², the lamprey-like AgnathaX¹³, and designs like the amphibious robotic turtle using soft joints¹⁴—exhibit impressive multimodal and environment-adaptive behaviors. However, these systems relying on rigid or passive materials (characterized by limited degrees of freedom, centralized control, tethered power sources, and restricted compliance) fundamentally differ from the autonomous soft robots discussed in this work, which feature infinite degrees of freedom, decentralized control, untethered power, and enhanced compliance—placing rigid-material-based systems outside the scope of our discussion.

On the other hand, for soft robots constructed entirely from soft materials and relying on built-in feedback loops to achieve autonomous motion, transitions between different self-sustained

motion modes under constant stimuli typically require human intervention. This includes changing the predesigned structural shape (*e.g.*, two-end constraints for undulation vs. one-end constraint for oscillation)⁶, manually adjusting the initial stimulus direction (*e.g.*, vertical scanning light for rightward propulsion vs. horizontal constant light for leftward propulsion)¹⁵, switching the stimulus type (*e.g.*, using light, heat, or solvent to trigger different locomotion modes)¹⁶, or tuning environmental constraints (*e.g.*, free space enabling spin vs. circular contact enabling orbit)¹⁷. We believe these examples of multimodal behavior do not qualify as *self-adaptive multimodal* performance due to the significant human intervention involved. Instead, they still represent single-mode operation within a fixed structure under the same type of stimulus.

To make it more precise, we have revised the Abstract as follows: “Autonomous soft robots with built-in feedback can achieve self-sustained motion under **constant**, untethered stimuli. However, **these systems** are constrained to single-mode actuation **within a fixed structure under the same type of stimulus** and require human intervention to switch modes, lacking the robust and efficient self-adaptation of living organisms in changing environments.” **And the relevant section in Introduction:** “Despite these advancements, existing examples of such physical intelligence-enabled soft robots remain in their infancy, constrained to single-mode actuation **for self-sustained motion within a fixed structure when triggered by the same type of stimulus**, and still dependent on human intervention (*e.g.*, reshaping the robot) to switch motion modes, as their response logic is mechanically pre-encoded into the material during fabrication¹⁸⁻²⁰.” **and** “Both modes serve important functions in biological kinematics and **have been accomplished using a monolithic material system, but each requires a distinct predesigned structure, with human intervention needed for switching or reconfiguration^{6,15}. To date, switching between different orders of motion modes (*e.g.*, undulation and oscillation) within a single structure under the same kind of stimulus remains unrealized—a key challenge in advancing physical intelligence—due to the difficulty of dynamically modulating** the inherent constraint of the structure.”

2. *In Fig. 1B, what’s the difference between the solid line and the dashed lines? For the undulation branch (similarly for the tracking branch), can a state exist that has a larger energy beyond the undulation-oscillation bifurcation point but still performs the undulation?*

Response:

Thank you for your highly constructive comments. As you pointed out, we also agree that the pink dashed line could be misleading, so we have revised it to a solid line in the updated Fig. 1B, as shown in Fig. R10A. In this schematic, the solid lines represent the three experimentally observed motion modes of the multimodal actuator, distinguished by different colors. The dashed lines indicate the hypothetical outputs that would occur without bifurcation,

corresponding to lower average energy-output modes in our case²¹. This aligns with the numerically simulated results shown in Fig. R10B. For example, at the first bifurcation point (3.728 W/cm²), the actuator would exhibit zero average kinetic energy if it remained in static tracking mode (blue dashed line), but due to bifurcation, it actually switches to the undulation mode with an increased slope of 0.0756 (purple line). Similarly, after the second bifurcation point, the value of the slope further jumps to 3.1602 (pink line), compared to 0.0756 in the undulation mode (purple dashed line).

Fig. R10: (A) Revised version of Figure 1B in Manuscript. (B) Average kinetic energy change across the two bifurcation points.

For the second question, within a given set of geometric parameters in a fixed environment, there is only one critical undulation–oscillation bifurcation point, as the instability and the photo-thermal-mechanical coupling are predetermined by the boundary conditions. This means a state with oscillation-level light input cannot coexist with or remain in a lower energy output mode like undulation beyond that critical point. As a result, tuning this critical point requires modifying the photo-thermal-mechanical coupling by adjusting either the geometric parameters (Fig. 3) or the environmental conditions (Fig. 4), as demonstrated in our subsequent experiments.

3. In Fig. 2B, do the pink and purple areas refer to the responses under the same thermal energy input? In line 120, the statement of “sufficient” seems unclear for the input condition.

Response:

Thank you for your detailed comments. The answer to your question is yes. In Fig. 2B, the pink and purple areas represent the temperature ranges under the same light intensity (3.728 W/cm²), based on the simulation results shown in Fig. S8A. To express this more clearly, we have

revised the caption of Fig. 2B as follows: “(B) Actuation stress of LCE/CSNP monolayer and bilayer under DMA isostrain condition (pink and purple regions refer to the temperature ranges of LCE/CSNP monolayer and bilayer, respectively, **under constant illumination**).”

For the second point, we have rewritten the relevant description to clarify the unclear point, as below: “In Region II, **although the bilayer exhibited a higher temperature under constant illumination, it** generated less than one-third of the stress (77.1-134.9 kPa) compared to the LCE/CSNP monolayer (362.9-475.6 kPa), **as indicated by the DMA results (Fig. 2B).**”

4. *In Fig. S6B, the immediate temperature drop is explained to be caused by the energy release in structural buckling. Are there any references showing a similar buckling-induced temperature drop, or is it caused by the side effects of fast motion or self-shadowing? Please provide more support information.*

Response:

Thank you for your constructive comments. We apologize for the misleading caused by the term “energy release.” The buckling-induced temperature drop is primarily attributed to convection cooling, which results from the rapid motion occurring at the moment of buckling (Fig. R11A). This sudden movement instantly enhances the convection cooling for heat dissipation, particularly from the top layer, leading to a momentary temperature drop. Following buckling, the top layer will continue to absorb light, initiating the bending-up motion and triggering the self-shadowing mechanism, as you noted (Fig. R11A). This self-shadowing mechanism induces periodic temperature drops, as shown in the temperature fluctuations observed between 0.3 s and the end of the cycle in Fig. R11B. In one cycle, the damping effect weakens, leading to a decrease in the oscillation speed of Region III. Consequently, the effect of convection cooling is reduced, resulting in a gradual increase in the overall temperature.

Specifically, based on the following equation, this drop is estimated to be on the order of 1 °C, consistent with the numerical simulation results (Fig. R11B, green line).

$$Q = h \times A \times (T_{LCE} - T_{air})$$

where Q is the heat transfer rate (W), h is the convective heat transfer coefficient (W/m²·K), and A is the surface area (3 mm × 25 mm).

$$\Delta T = \frac{Q \times t}{C}$$

Meanwhile, the overall temperature variation remains within the same order of magnitude as other similar photothermal cantilever-structured oscillators—fluctuating within a range of several degrees Celsius (Fig. R11D, corresponding to Fig. 1D in ⁸; Fig. R11E, corresponding to Fig. S17 in ⁶).

Fig. R11: (A) Schematic illustration of buckling-induced undulation: initial buckling is followed by upward bending that triggers the self-shadowing mechanism. Temperature variation in the undulation cycle obtained from (B) simulation and (C) IR camera measurements. Temperature–deformation profiles of self-shadowing cantilever beams based on (D) AuNR/hydrogel and (E) LCE/PDMS.

To describe this process more clearly, we note that the temperature shown in Fig. S8B (updated as Fig. R11C) represents the peak temperature captured by the IR camera. As such, it reflects the periodic temperature drop induced by buckling, followed by a gradual increase—consistent with the maximum temperature profile obtained from the numerical simulation in Fig. R11B. Accordingly, we have rewritten the relevant section in Manuscript as follows: “IR camera imaging and simulation modeling revealed a sudden temperature drop at each buckling moment, followed by a gradual temperature rise that initiates the next cycle (Fig. 2A right and Supplementary Fig. 8B).” and revised the caption of Fig. S6B (now Fig. S8B) as: “**Fig. S8. Temperature variation in the instability-induced undulation cycle.** (A) As simulation results suggest, for each cycle, the top layer has a higher temperature than the bottom layer during the buckling since it is heated directly. During the damping, the two temperature differences decrease to thermal equilibrium. (B) As analyzed from IR camera data (capturing the maximum temperature of the trilayer), buckling caused a rapid temperature drop due to strong convection cooling, followed by the excitation of Region III into oscillation via the self-shadowing mechanism. Over time, as damping weakened and oscillation speed decreased, the resulting reduction in convection cooling led to a gradual increase in overall temperature.”

5. In the caption of Fig. S15, double-check the values of the two groups of parameters.

Response:

Thanks for your detailed review. As you pointed out, there was indeed a mistake, and the value of the second X/L in the original caption should be 0.10. To better describe Fig. S15 (now Fig. S18) with a clearer definition, we have revised the caption as follows: “**Fig. S18. Comparison of the relationship between K and K_0 .** The red hollow symbols represent states where $K < K_0$, indicating that the bending tendency dominates and the trilayer remains in the tracking or oscillation state. In contrast, the solid symbols indicate $K > K_0$, where the buckling tendency dominates, enabling the undulation. For X/L = 0.20 and d/L = 0.12, K can exceed the critical K_0 (blue dotted line), leading to the undulation. In contrast, for groups where K consistently remains below K_0 (e.g., X/L = 0.10 and d/L = 0.12; purple dotted line), the systems are limited to exhibiting only tracking and oscillation.”

6. When swimming in viscous fluid with a low Reynolds number, microorganisms use different but more efficient locomotion compared to the fish-like oscillation. In terms of speed or efficiency, can the swimming robot benefit from the automatic switch of motion modes? Please clarify the motivation for mode-switching.

Response:

Fig. R12: (A) Comparison of kinetic energy between undulations and oscillations. (B) Comparison of peak and average energy conversion efficiencies between undulation and oscillation modes.

Thank you for your valuable comments. The motivation for mode-switching in our design was inspired by the behavioral strategy of *Gymnura micrura*, which employs undulatory swimming at lower speeds to enhance instantaneous output and maneuverability during predation or reproduction, and transitions to oscillatory swimming for high-speed cruising⁵. This closely parallels the behavior of our multimodal actuator. Based on modeling results near the

undulation–oscillation transition point (Fig. R12A), we calculated both the peak and average output efficiencies of undulation (5.414 W/cm²) and oscillation (5.494 W/cm²) using the following equations, and found that undulation exhibits a peak output more than twice that of oscillation, whereas oscillation yields an average output approximately 1.5 times higher than that of undulation (Fig. R12B).

$$E_{\text{input}} = I \times A \times t$$

$$\eta_{\text{peak}} = \frac{\text{Max(Kinetic energy)}}{E_{\text{input}}}$$

$$\eta_{\text{avg}} = \frac{\text{Ave(Kinetic energy)}}{E_{\text{input}}}$$

Moreover, the mode-switching ability enables our actuator to maintain continuous locomotion after encountering physical contact (Fig. 4E, Fig. S23, and Movie S11) Specifically, the asymmetric actuator achieved a speed of 0.438 mm/s in free movement and 0.140 mm/s under the physical contact, which maintains sufficient for self-sustained locomotion. In contrast, the symmetric trilayer with only the oscillation mode could reach a higher speed of 1.003 mm/s without constraint under a higher light intensity (6.723 W/cm²), but its speed dropped sharply to 0.007 mm/s with the same contact, resulting in negligible forward movement. This performance is also similar to the behavior of *Gymnura micrura*, where low-speed undulatory swimming is preferred in confined spaces to improve maneuverability and control. To better clarify the motivation for mode-switching, the relevant descriptions in the Introduction and Results sections have been revised as follows: “The availability of motion switching creates a dynamic photo-thermo-mechanical feedback system with the potential for multimodal adaptation under varying conditions (e.g., light intensity, fluid viscosity, surrounding temperature, and physical constraint) controlled by the energy state. This leads to distinct performance outputs—analogue to the behavioral strategy of *Gymnura micrura*—with higher peak output during undulation but higher average output during oscillation, allowing for continuous locomotion across diverse environments”, “Furthermore, we observe bursts of energy output every cycle in undulation mode, while a higher average kinetic energy output is presented in oscillation despite lower energy maxima near the undulation–oscillation bifurcation point (Fig. 3F and Supplementary Text 2.3). Specifically, the peak output efficiency of undulation is more than twice that of oscillation, while its average output efficiency is only 0.681 times that of oscillation (Supplementary Fig. 19)—a trade-off akin to the behavioral strategy of *Gymnura micrura*, which employs undulatory swimming at lower speeds to enhance instantaneous output and maneuverability during predation or reproduction, and shifts to oscillatory swimming under high-speed cruising⁵” and “This undulation of the asymmetric actuator indicated its ability to achieve the higher maneuverability to navigate obstacles under

lower energy output, paralleling the undulatory locomotion of *Gymnura micrura* that adopts this strategy at low speeds to improve controllability in confined spaces.”

Meanwhile, we have integrated Fig. R12B into the Supplementary Materials as Fig. S19, as shown below: “**Fig. S19. Comparison of peak and average energy conversion efficiencies under undulation and oscillation modes.** The total input energy was calculated as

$$E_{\text{input}} = I \times A \times t$$

where I was the light intensity, A was the illuminated area (a rectangular spot of 3.5 mm × 4.5 mm), and $t = 1.1$ s was chosen to ensure at least one full motion cycle (Fig. 3F). Accordingly, the peak efficiency (η_{peak}) was defined as

$$\eta_{\text{peak}} = \frac{\text{Max(Kinetic energy)}}{E_{\text{input}}}$$

and the average efficiency (η_{avg}) as

$$\eta_{\text{avg}} = \frac{\text{Ave(Kinetic energy)}}{E_{\text{input}}}$$

where the kinetic energy was calculated based on the simulated modeling results.”

7. In Fig. S17D, the vertical coordinate may infer that the tracking or undulation can have a higher output energy or speed than oscillation, which is a bit confusing.

Response:

Fig. R13: (A) Revised version of Fig. S21D in Supplementary Materials.

Thank you for your careful review and sorry for the misleading. We have revised Fig. S17D (now Fig. S21D), shown as Fig. R13. As you correctly noted, increasing the viscosity enhances the damping resistance, which leads to a reduction in output and a sequential transition in motion modes—from oscillation to undulation, and ultimately to tracking.

8. In Fig. 2C, the data in the horizontal coordinate axis are a little crowded and unclear.

Response:

Fig. R14: (A) Revised version of Fig. 2C in Manuscript.

Thanks for your detailed suggestion. Following your advice, we adjusted the font size and spacing of the horizontal axis of Fig. 2C to eliminate overlap to present in a clearer and more readable way (Fig. R14).

9. In lines 215-216, it would be clearer to clarify that Fig. 3E is the numerically simulated result.

Response:

Thank you for your comments. We have revised the relevant description to emphasize that the results shown in Fig. 3E are based on numerically simulated data, as follows: “Such numerically simulated results update the map to Fig. 3E, where the white region indicates the occurrence of undulation—the key condition for achieving two bifurcations overserved in experiments (Fig. 3A).”

10. In line 279, it should be “PDMS” rather than “PMDS”.

Response:

Thank you for your careful reading. We have corrected the typo in the relevant section as follows: “Additionally, the periodic vacancy changes in the bow-shaped structure during buckling-induced undulation make the asymmetric actuator promising for developing triboelectric nanogenerators (TENG) by adopting PTFE/Pt and PDMS/Pt as inner layers.”

Reference

- 5 Rosenberger, L. J. Pectoral Fin Locomotion in Batoid Fishes: Undulation Versus Oscillation. *Journal of Experimental Biology* **204**, 379-394 (2001).
- 6 Zhao, Y. *et al.* Sunlight-powered self-excited oscillators for sustainable autonomous soft robotics. *Science Robotics* **8**, eadf4753 (2023).
- 8 Zhao, Y. *et al.* Soft phototactic swimmer based on self-sustained hydrogel oscillator. *Science Robotics* **4**, eaax7112 (2019).
- 12 Bouman, A. *et al.* in *2020 IEEE/RSJ International Conference on Intelligent Robots and Systems (IROS)*. page 2518-2525 (IEEE).
- 13 Thandiackal, R. *et al.* Emergence of robust self-organized undulatory swimming based on local hydrodynamic force sensing. *Science Robotics* **6**, eabf6354 (2021).
- 14 Baines, R. *et al.* Multi-environment robotic transitions through adaptive morphogenesis. *Nature* **610**, 283-289 (2022).
- 15 Shahsavan, H. *et al.* Bioinspired underwater locomotion of light-driven liquid crystal gels. *Proceedings of the National Academy of Sciences* **117**, 5125-5133 (2020).
- 16 He, Q. *et al.* A modular strategy for distributed, embodied control of electronics-free soft robots. *Science advances* **9**, eade9247 (2023).
- 17 Qi, F. *et al.* Defected twisted ring topology for autonomous periodic flip–spin–orbit soft robot. *Proceedings of the National Academy of Sciences* **121**, e2312680121 (2024).
- 18 Li, Z., Myung, N. V. & Yin, Y. Light-powered soft steam engines for self-adaptive oscillation and biomimetic swimming. *Science Robotics* **6**, eabi4523 (2021).
- 19 Park, S.-J. *et al.* Phototactic guidance of a tissue-engineered soft-robotic ray. *Science* **353**, 158-162 (2016).
- 20 Djellouli, A. *et al.* Shell buckling for programmable metafluids. *Nature* **628**, 545-550 (2024).
- 21 Ling, F., Guo, H. & Kanso, E. Instability-driven oscillations of elastic microfilaments. *Journal of The Royal Society Interface* **15**, 20180594 (2018).

Reviewer #4

Response:

We sincerely appreciate your time and valuable feedback.